# ARGONAUTE10 controls cell fate specification and formative cell divisions in the Arabidopsis root

Nabila El Arbi [ID] [1,3,5], Ann-Kathrin Schürholz[1,4,5], Marlene U Handl [ID] [1,2,5], Alexei Schiffner [ID] [1,2], Inés Hidalgo Prados [ID] [1], Liese Schnurbusch [ID] [2], Christian Wenzl[1], Xin'Ai Zhao [ID] [1], Jian Zeng[1], Jan U Lohmann [ID] [1] & Sebastian Wolf [ID] [1,2✉]

## Abstract

A key question in plant biology is how oriented cell divisions are integrated with patterning mechanisms to generate organs with adequate cell type allocation. In the root vasculature, a gradient of miRNA165/6 controls the abundance of HD-ZIP III transcription factors, which in turn control cell fate and spatially restrict vascular cell proliferation to specific cells. Here, we show that vascular development requires the presence of ARGONAUTE10, which is thought to sequester miRNA165/6 and protect HD-ZIP III transcripts from degradation. Our results suggest that the miR165/6-AGO10-HDZIP III module acts by buffering cytokinin responses and restricting xylem differentiation. Mutants of AGO10 show faster growth rates and strongly enhanced survival under severe drought conditions. However, this superior performance is offset by markedly increased variation and phenotypic plasticity in sub-optimal carbon supply conditions. Thus, AGO10 is required for the control of formative cell division and coordination of robust cell fate specification of the vasculature, while altering its expression provides a means to adjust phenotypic plasticity.

**Keywords** Plant Development; Cell Fate; Robustness; Cytokinin; Arabidopsis
**Subject Categories** Development; Plant Biology; RNA Biology

See also: S Mirlohi et al

## Introduction

One of the key evolutionary innovations that allowed plants to dominate the vast majority of terrestrial habitats (Bar-On et al, 2018) was the development of vascular systems comprised of xylem and phloem cells which allowed water and nutrient transport over long distances (Lucas et al, 2013). In the model plant *Arabidopsis thaliana*, precursor cells giving rise to the primary root vasculature are specified in the embryo (De Rybel et al, 2016), but undergo periclinal or radial (i.e. formative) divisions post-embryonically to give rise to additional cell files and thus complete pattern formation.

Cell fate determination in plants is mainly based on positional information, rather than cell lineage (Efroni et al, 2016; Kidner et al, 2000; Berger et al, 1998; van den Berg et al, 1997, 1995). For establishing the positional cues involved in vascular cell fate patterning, the interplay of auxin and cytokinin (CK) hormone signalling pathways is crucial (Müller and Sheen, 2008; Hamann et al, 2002; Hardtke and Berleth, 1998). Five xylem precursor cells arranged in a single axis are characterized by high auxin response, whereas two phloem poles arranged perpendicular to the xylem and the intervening procambium cells, are characterized by high CK response (Appendix Fig. S1A). Importantly, CK signalling is required for formative cell division in the stele, as shown by the reduced number of vascular cell files in plants lacking CK receptors or the ARABIDOPSIS RESPONSE REGULATOR (ARR) TFs that mediate CK responses (Argyros et al, 2008; Ishida et al, 2008; Yokoyama et al, 2006; Mähönen et al, 2000; Mähönen, 2006). At least in part, CK in the stele acts through controlling the expression of an array of DNA-BINDING WITH ONE FINGER (DOF) transcription factors that promote formative cell divisions (Miyashima et al, 2019; Smet et al, 2019). Another group of TFs, the class III HOMEODOMAIN LEUCINE-ZIPPER (HD-ZIP III) family proteins, counteract a subset of the DOF factors and inhibits periclinal cell divisions (Miyashima et al, 2019; Carlsbecker et al, 2010). The accumulation of HD-ZIP III proteins, such as PHABULOSA (PHB/AtHB-14), PHAVOLUTA (PHV/AtHB-9), and CORONA (CAN/AtHB-15), is restricted to the centre of the stele by an intricate regulatory mechanism involving the non-cell autonomous action of miRNAs in the miR165/166 family (Carlsbecker et al, 2010; Rhoades et al, 2002). Transcription of the *MIR165/6* genes in the root occurs specifically in the endodermis, which surrounds the vascular cylinder. Upon cell-to-cell movement through plasmodesmata, miR165/6, or a precursor thereof, is thought to form a gradient with a minimum in the centre of the stele. Presence of miR165/6 leads to the degradation (slicing)

[1]Centre for Organismal Studies Heidelberg, Heidelberg University, Im Neuenheimer Feld 230, 69120 Heidelberg, Germany. [2]Center for Plant Molecular Biology (ZMBP), University of Tübingen, Auf der Morgenstelle 32, 72076 Tübingen, Germany. [3]Present address: Department of Plant Physiology, Umea Plant Science Centre, Umea, Sweden. [4]Present address: Corden Pharma, Heidelberg, Germany. [5]These authors contributed equally: Nabila El Arbi, Ann-Kathrin Schürholz, Marlene U Handl. ✉E-mail: sebastian.wolf@zmbp.uni-tuebingen.de

of *HD-ZIP III* transcripts through ARGONAUTE1 (AGO1) (Miyashima et al, 2019; Carlsbecker et al, 2010; Kidner and Martienssen, 2004), restricting the transcription factors (TFs) to the centre of the stele (Carlsbecker et al, 2010). Due to the presence of HD-ZIP III proteins, the central cells of the stele generally only divide anticlinally, but not periclinally (Miyashima et al, 2019). The closest relative of AGO1, ARGONAUTE10 (AGO10, also known as ZWILLE and PINHEAD (Moussian et al, 1998; McConnell and Barton, 1995; Jürgens et al, 1994)) has been shown to compete with AGO1 for miRNA165/6 (Zhu et al, 2011; Liu et al, 2009; Mallory et al, 2009; Lynn et al, 1999) and to protect HD-ZIP III transcripts from AGO1-mediated degradation in the shoot apical meristem (SAM) (Liu et al, 2009). AGO10 mutants have been described to display embryonic and post-embryonic defects in the shoot meristems that can lead to stem cell differentiation and meristem termination (Moussian et al, 1998; Zhu et al, 2011; Liu et al, 2009; Lynn et al, 1999). However, these phenotypes are only observed in a minority of Arabidopsis ecotypes with a restricted geographical distribution (Mallory et al, 2009; Tucker et al, 2013; Takeda et al, 2008). Here, we show that AGO10 is required for the control of formative cell divisions in the centre of the root stele, since its loss leads to ectopic xylem strands, increased procambial cell number, and enhanced root growth. AGO10 is mostly expressed in the centre of the meristematic stele, consistent with a role in protecting HD-ZIP III transcripts from miRNA165/6-mediated degradation. Furthermore, AGO10-guarded HD-ZIP III activity buffers CK responses in the stele to coordinate oriented cell divisions with patterning. Our results suggest that AGO10 is required for maintenance of an instructive miRNA165/6 gradient, presumably by adapting steepness of the gradient to the range required by the cellular layout of the root. Potentially as a result of increased vascular cell number, AGO10 mutants outperform the wild type under water-limiting conditions. However, under other conditions, AGO10 is critically required for phenotypic robustness, underlining its essential role in miRNA gradient shaping.

## Results

### A mutant with ectopic xylem formation in the root vasculature

Arabidopsis roots display a stereotypical patterning of primary xylem in one axis, with two protoxylem cells at the periphery and three, in rare cases four, metaxylem cells in the centre of the axis (Appendix Fig. S1A). With the aim to study a gene we hypothesized to be involved in vascular development (*RECEPTOR-LIKE PROTEIN4*, At1g28340), we generated mutant lines in *Arabidopsis thaliana* Col-0 plants using CRISPR/Cas9. We discovered xylem phenotypes in two independent lines that showed frame shift mutations in the target gene, but the phenotype segregated away from the *rlp4* mutation, indicating that the latter was not causative for the observed effects on xylem. We studied the novel mutant that we named *the show must go on 1* (*sgo1*), after outcrossing of the Cas9 transgene and backcrossing to wild type Col-0. The recessive (Appendix Fig. S1B) *sgo1* mutants showed an increased number of cells with lignified protoxylem-like secondary cell wall thickenings (Fig. 1A,B). These cells were always found at the periphery of the stele, indicating that general xylem patterning remained intact in

the mutant. However, some of these additional protoxylem cells were placed outside of the xylem axis (Fig. 1A). The increase in apparent protoxylem cells did not occur at the expense of metaxylem cells. In contrast, *sgo1* showed slightly increased metaxylem cell number, with infrequent occurrence of lignified cells outside of the xylem axis (Fig. 1A,C,D).

To assess whether this increased xylem cell number in *sgo1* is due to cell fate changes in the procambium (Holzwart et al, 2018) or rather is the consequence of increased xylem precursor cell division activity, we generated optical cross sections through the stele. We quantified the cell number close to the stem cell region, 15 μm and 22 μm above the quiescent centre (QC), and 150 μm above the QC (Fig. 1E), where the final stele patterning has been established (Miyashima et al, 2019). At all positions, *sgo1* cell number was significantly increased compared to the Col-0 wild type (Fig. 1E), indicating more formative divisions in the root meristem. To test whether increased number of xylem precursor cells in *sgo1* could be due to loss of quiescence in xylem precursor and inner procambial cells (Miyashima et al, 2019), we quantified cell division activity by monitoring incorporation of the thymidine analogue 5-ethynyl-2′-deoxyuridine (EdU) into S-phase nuclei (Kotogány et al, 2010). After a 2-hour EdU treatment and subsequent click chemistry addition of a fluorophore, labelled nuclei were dispersed throughout the root meristem (Fig. 2A), with *sgo1* displaying slightly higher numbers of S-phase nuclei (Fig. 2B). We then followed a pulse-chase strategy incorporating an EdU-free period of 6 h following a 40-minute EdU pulse (Fig. 2C; Movies EV1, EV2). We reasoned that this should allow a majority of S-phase cells that acquired EdU during the pulse to proceed through the G2- and M-phases during the chase period. Accordingly, we observed fluorescence occasionally in mitotic structures (Fig. 2C, arrow) and predominantly in paired nuclei (Fig. 2C, arrowheads). As expected, the majority of fluorescently labelled and paired cells were oriented parallel to the root long axis, indicating a recent anticlinal division. We focused on pairings periclinal to the root long axis, since these could be derived from recent formative cell divisions. In accordance with previous observations regarding the spatial distribution of periclinal divisions (Miyashima et al, 2019), periclinal pairs of fluorescent nuclei were restricted to positions adjacent to the pericycle in the WT (Fig. 2D,F). In sharp contrast, *sgo1* roots showed periclinal pairs throughout the stele (Fig. 2E,F). Notably, the resulting enhanced radial growth of the root did not impede longitudinal root growth; in fact, *sgo1* mutants exhibited longer roots than the wild type (Fig. 2G), suggesting that the control of SGO1 over cell divisions potentially extends beyond the stele. In summary, SGO1 is required to suppress cell division activity in the centre of the stele, such that increased overall cell number in the mutant could be causative for the observed increase in differentiated xylem cells (Fig. 1A–D).

### Inner vascular cells ectopically divide and maintain lineage-specific cell identity in *sgo1*

To determine whether *sgo1* roots exhibit an increase in meristematic xylem precursor cells, we crossed the mutant with fluorescent marker lines and quantified reporter-positive cells throughout the meristem. We quantified protoxylem precursor cells as indicated by reporter fluorescence driven by the AHP6 promoter (Mähönen, 2006) and observed an enlarged *pAHP6:erGFP* expression domain

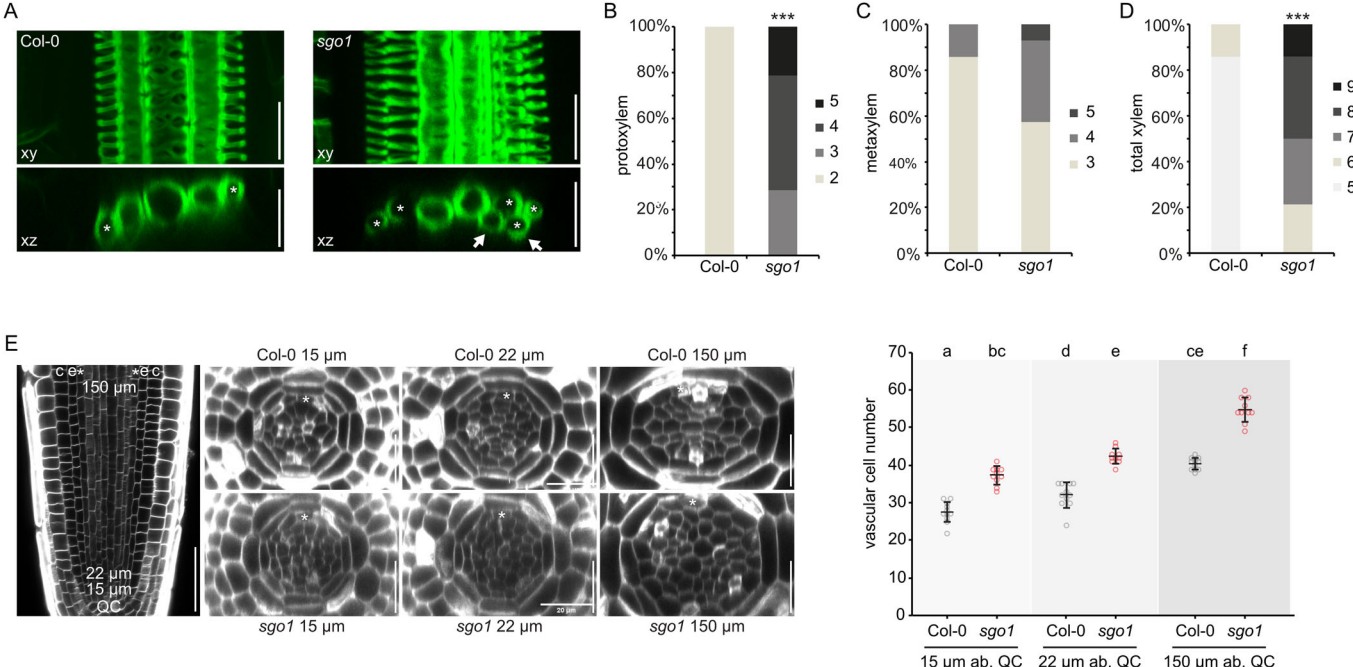

**Figure 1. *sgo1* displays ectopic xylem formation and increased formative divisions in the primary root.**

(A) Lignified xylem cells in Col-0 and *sgo1*. Upper panels are xy projections of a confocal stack, lower panels optical xz sections through the same stack. Asterisks denote cells with protoxylem differentiation, arrows point to ectopic xylem strands. Scale bar = 25 μm. (B–D) Frequency of roots with the indicated number of protoxylem (B), metaxylem (C) or total xylem (D) cells in Col-0 and *sgo1*. *n* = 14 biological replicates. (E) Quantification of vascular cell number in cross section of confocal stacks at 15 μm, 22 μm, and 150 μm distance from the quiescent centre (QC) cells. Panels on the left show representative Col-0 and *sgo1* meristems. c = cortex, e = endodermis, asterisk = pericycle. Data information: In (B–D), asterisks indicate statistically significant difference from Col-0 based on Mann–Whitney U test (\*\*\**P* < 0.001). In (E), graph depicts means ± s.d. and individual data points (*n* = 9–11 biological replicates). Letters in graph indicate statistically significant differences based on Tukey's post hoc test after one-way ANOVA. Scale bar in left panel = 50 μm, other scale bars = 25 μm. Source data are available online for this figure.

150 μm above QC (Fig. 3A,B). This enlargement occurred within the xylem axis, but also occasionally included cells in the procambium domain, consistent with what was observed in fully differentiated xylem cells. The auxin response marker *DR5v2:YFPer* (Ma et al, 2019) showed an enlarged expression domain in *sgo1* already at 15 μm above the QC (Fig. 3C). It is noteworthy that in the wild type, *DR5v2:YFPer*-derived signal was not strictly correlated with xylem precursor cells directly above the QC but showed quite variable and widespread expression, which stabilized and constricted to the five xylem precursor cells as patterning was completed at 150 μm distance from the QC (Fig. 3C). In *sgo1*, the *DR5v2:YFPer* expression pattern remained variable at all positions and was markedly expanded compared to the wild-type background (Fig. 3C).

To quantify the number of cells with xylem identity in the meristem, we analysed nuclear GFP driven by the TMO5 promoter (Schlereth et al, 2010). *TMO5* expression is largely controlled by auxin, however, the reporter exhibits a more restricted activity compared to *DR5v2:YFPer*. In most wild-type roots, *TMO5:NLS-3xGFP* was confined to five cells (Fig. 3D), consistent with faithful representation of xylem precursor cells. In contrast, the number of *TMO5:NLS-3xGFP*-positive cells in *sgo1* roots was substantially increased throughout the meristem (Fig. 3D; Movies EV3, EV4). Quantification of pTMO5-positive cell files that were the apparent result of ectopic xylem precursor cell divisions. i.e. files located outside of the xylem axis, revealed on average more than three of

those cell files in *sgo1* (Fig. 3E). In some cases, these ectopic files could be traced back to a likely common precursor cell (Fig. 3E, arrow). Consistent with expectations, ectopic xylem precursor cell files were virtually absent in the wild-type background. Likewise, and in contrast to the Col-0 wild type, *sgo1* roots frequently showed additional cell files resulting from formative divisions in inner procambium cells directly adjacent to *pTMO5:NLS-3xGFP*-positive cells (Fig. 3F). Notably, the increased formative division in *sgo1* did not occur randomly but led to continuous strands of TMO5-positive precursor cells, even when the extra strands appeared displaced from the xylem axis. However, the length of these ectopic strands varied. We could observe ectopic TMO5-positive cell files that spanned the entire meristem (Fig. 3G) as well as the initiation of ectopic strands (Fig. 3E, arrow) and beginning displacement from the meristem (Fig. 3F, arrows). This is consistent with our observations in the mature part of the root, where additional xylem files are also largely continuous (Fig. 1). In summary, SGO1 is involved in the control of stele cell number and xylem proliferation.

## SGO1 encodes AGO10

Through a combination of bulked segregant analysis and whole genome sequencing, we identified a single nucleotide polymorphism (G1333A counting from ATG) in the *AGO10* coding sequence in *sgo1*, predicted to incur an E-to-K amino acid substitution at position 445 in the AGO10 protein. To determine whether

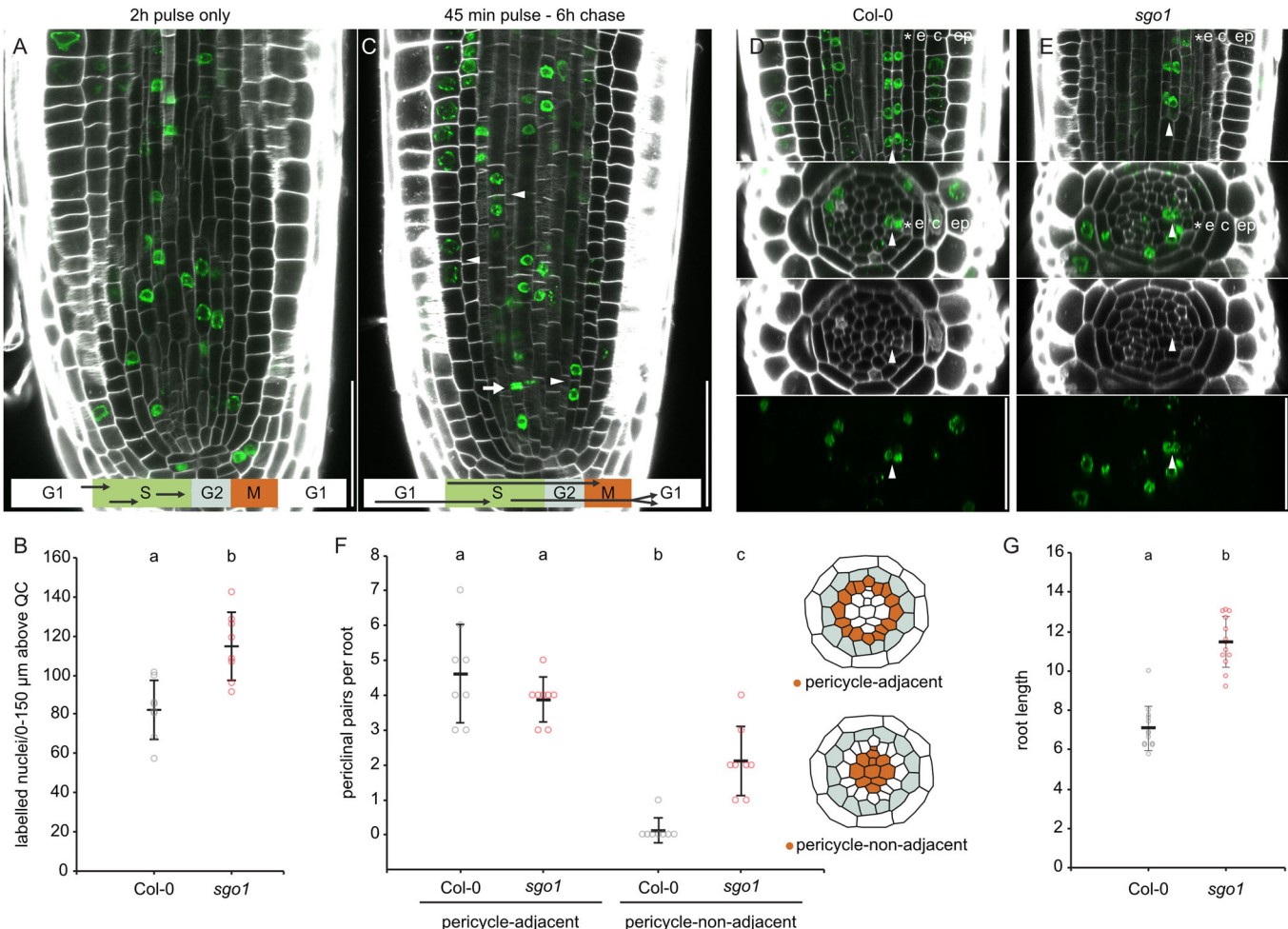

**Figure 2.** *sgo1* shows enhanced periclinal cell division activity in the root meristem.

(A) Exemplary image of a wild-type (Col-0) root with calcofluor white-stained cell walls and Alexa-488-labelled S-phase nuclei after 2 h of EdU exposure. Arrows in scheme indicate individual labelled cells. (B) Quantification of EdU-labelled nuclei in the vascular cylinder at 0–150 μm distance from the QC. (C) Exemplary image of a Col-0 root after 45 min of EdU feeding and a 6-hour chase period without EdU. Note labelling of mitotic structures (arrows) and nuclei pairs (arrowheads), likely descendants of cells acquiring EdU during previous S-phase. Arrows in scheme indicate individual labelled cells. (D, E) Representative confocal stacks of a Col-0 (D) and *sgo1* (E) meristem showing pairing of nuclei indicative of periclinal cell divisions. Image series shows same periclinal pairs in xy section (upper panel) and xz cross sections (lower three panels) as composite image, cell wall and Alexa-488 channel, respectively. (F) Quantification of periclinal nuclei pairs in the vascular cylinder (excluding pericycle) at 0–150 μm distance from the QC. Cells adjacent and non-adjacent to the pericycle are discriminated. (G) Root length of Col-0 and *sgo1* seedlings six days after germination. Data information: (B), (F), and (G) denote means ± s.d., individual data points are indicated. $n = 8$ biological replicates for (B) and (F), $n = 13$ biological replicates for (G). Letters in graph indicate statistically significant differences based on t-test (B, G) or Tukey's post hoc test after one-way ANOVA (F). Scale bars = 50 μm. Source data are available online for this figure.

mutation of AGO10 was causative for the *sgo1* phenotype, we analysed the recessive AGO10 loss-of-function mutant *zll-3* (Moussian et al, 1998) in the Landsberg *erecta* (L*er*) background. *zll-3* roots displayed additional differentiated xylem files as well as increased number of meristematic cell files, very similar to the *sgo1* mutant (Fig. EV1A–C). Consistent with this, F1 trans-heterozygotes resulting from a cross of *sgo1* with *zll-3* showed the *sgo1*/*zll-3* xylem phenotype, suggesting that the *ago10* SNP in *sgo1* is the causative mutation (Fig. EV1D). To characterize the *sgo1* allele further, we used western analysis employing an antibody raised against AGO10 (Iki et al, 2018) with extracts from Col-0, *sgo1*, and *ago10-1*, which is a T-DNA insertion line in the Col-0 background (Takeda et al, 2008). Whereas AGO10 protein was undetectable in *ago10-1*, *sgo1* showed slightly elevated AGO10 levels compared to the wild type, which could be due to the increased amount of AGO10 positive cells in the mutant or feedback-mediated upregulation (Fig. EV1D). Despite this discrepancy in AGO10 protein amount, the two mutant alleles showed a meristematic cell number phenotype indistinguishable from each other (Fig. 4A), suggesting that *sgo1* behaves as an AGO10 loss-of-function mutant consistent with mutation of the same amino acid in the strong *zll-7* allele (Moussian et al, 1998). In summary, *SGO1* encodes AGO10, and *sgo1*, *ago10-1*, and the L*er* allele *zll-3* are phenotypically indistinguishable in terms of vascular patterning defects. Similar defects in *ago10-1* are also reported in the accompanying study by Mirlohi et al (2024), which also describes

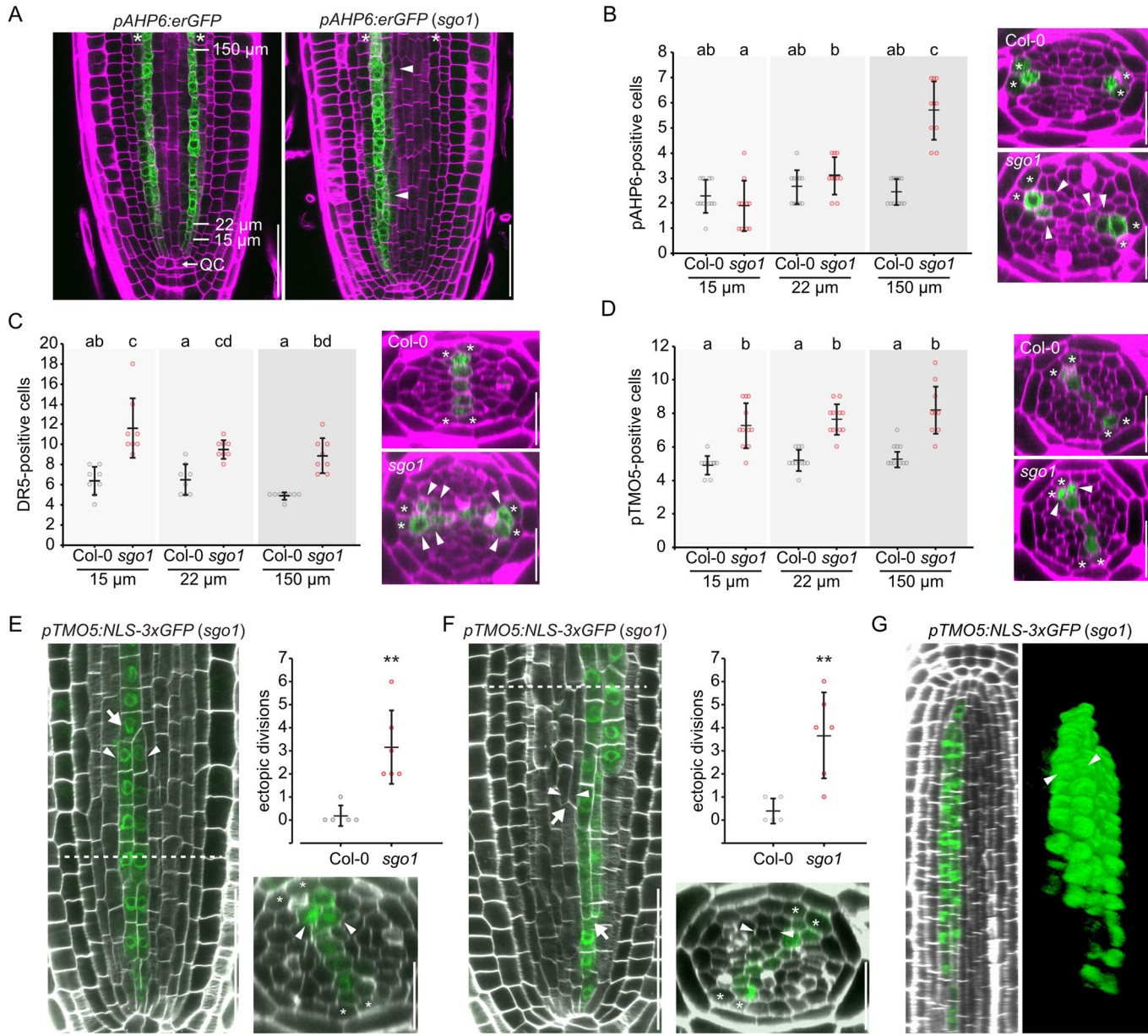

**Figure 3. Inner vascular cells ectopically divide and maintain lineage-specific cell identity in *sgo1*.**

(A) Expression of the *pAHP6:GFPer* marker is expanded inward in *sgo1*, while expression in pericycle cells is reduced. Asterisks = pericycle. (B) Quantification of *pAHP6:GFPer*-positive xylem cells in cross sections of confocal stacks of WT and *sgo1* at the indicated distances from the QC. n = 10–11 biological replicates. (C) Quantification of *pDR5v2:YFPer*-positive xylem cells in cross sections of confocal stacks of Col-0 and *sgo1*. n = 8 biological replicates. (D) Quantification of *pTMO5:NLS-3xGFP*-positive xylem cells in cross sections of confocal stacks of Col-0 and *sgo1*. n = 10–11 biological replicates. (E) *pTMO5:NLS-3xGFP*-positive xylem precursor cells ectopically divide in *sgo1* but not Col-0. Arrow points to initiation of a continuous ectopic cell file, i.e the likely common precursor cell, whereas arrowheads indicate the initiation of an ectopic cell file. n = 6 biological replicates. (F) Inner procambial cells adjacent to *pTMO5:NLS-3xGFP*-labelled xylem precursor cells ectopically divide in *sgo1*. Arrow points to a continuous ectopic cell file being displaced from the meristem. Dashed lines indicate position of xz section. n = 6 biological replicates. (G) pTMO5:GFP-labelled ectopic cell files in a longitudinal cross section (left panel) and maximum projection of the same confocal stack (right panel). Arrowheads indicate two xylem precursor cell files outside the xylem axis. Data information: graphs in (B–D) depict means ± s.d., individual data points are indicated. Letters indicate statistically significant differences based on Tukey's post hoc test after one-way ANOVA. Graphs in (E) and (F) display mean of ectopic divisions per root (0–150 µm distance from QC) ± s.d., individual data points are indicated. Asterisks in graphs indicate statistically significant differences based on student's t-test (**P < 0.01). Asterisks in micrographs denote pericycle cells at xylem poles, arrowheads indicate ectopic marker expression (A–E) or ectopic periclinal division (F). Scale bar in xy sections = 50 µm, in xy sections = 25 µm. Source data are available online for this figure.

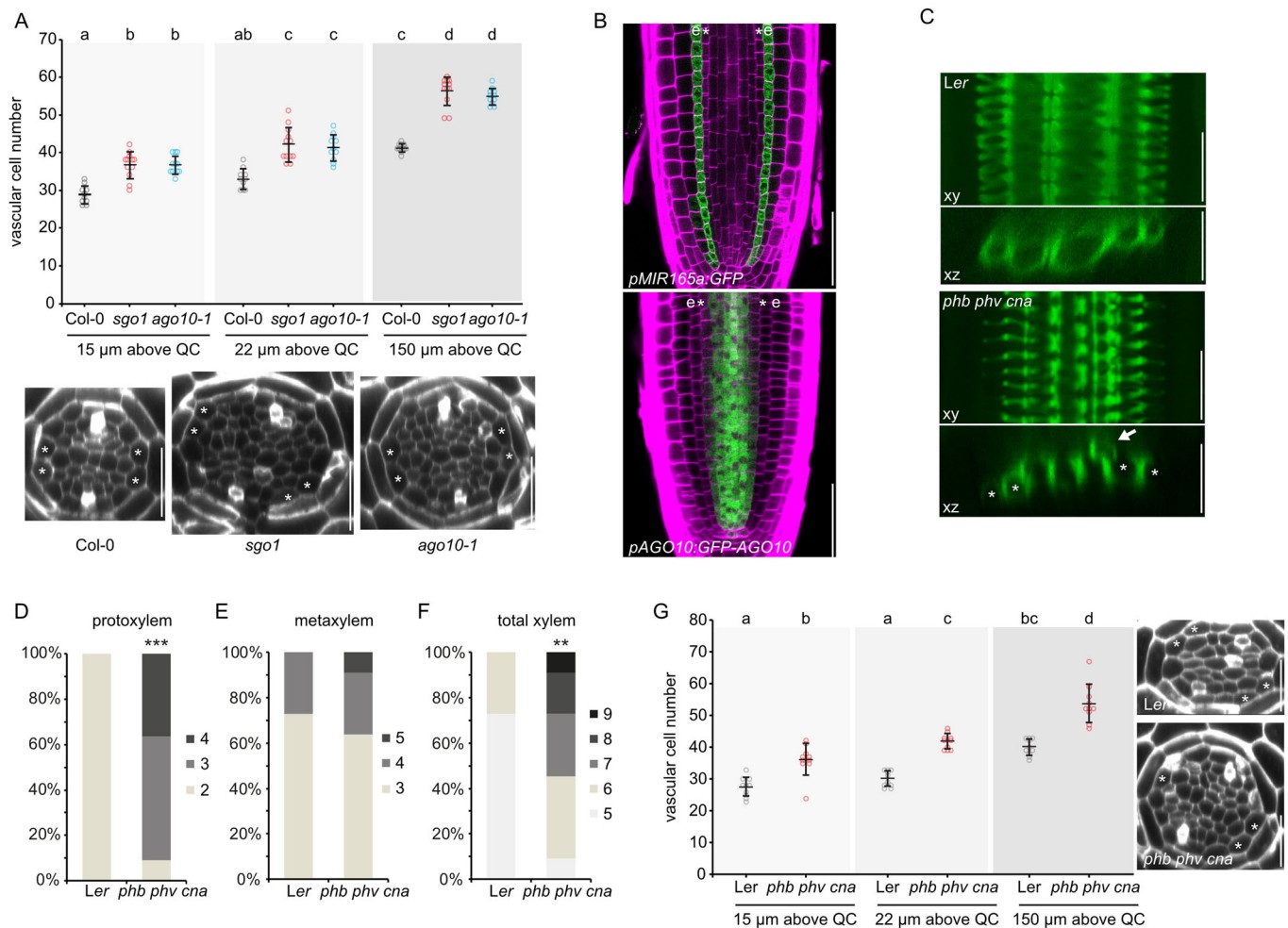

**Figure 4. AGO10 is required for HD-ZIP III-mediated vascular patterning.**

(A) Quantification of vascular cell number in cross section of confocal stacks. Scale bar = 50 μm. (B) Expression of *pMIR165a:GFP* (in Col-0 background) and *pAGO10:GFP::AGO10 (ago10-1)* reporters with near wild-type expression levels (Mirlohi et al, 2024) in the root meristem. e = endodermis, asterisks = pericycle. Scale bar = 50 μm. (C) Staining of lignified xylem cells. Upper panels are xy projections of a confocal stack, lower panels depict xz sections through the same stack. Scale bar = 25 μm, arrow points to ectopic xylem strand. (D–F) Frequency of roots with the indicated number of protoxylem (D), metaxylem (E) or total xylem (F) cells in Ler and *phb phv cna*. (G) Quantification of vascular cell number in cross sections of confocal stacks. Asterisks = pericycle. Data information: In (A), graph depicts means ± s.d. and individual data (n = 10–12 biological replicates). Letters in graph indicate statistically significant differences based on Tukey's post hoc test after one-way ANOVA. In (D–F), asterisks indicate statistically significant difference based on Mann–Whitney U test (***P < 0.001, **P < 0.01). n = 11 biological replicates. In (G), graph depicts means ± s.d. and individual data points (n = 10 biological replicates). Letters indicate statistically significant differences based on Tukey's post hoc test after one-way ANOVA. Source data are available online for this figure.

how over-expressing AGO10 in its cognate root expression domain yields xylem defects essentially mirroring those of *ago10-1*. Thus, in contrast to the ecotype-specific contribution of AGO10 to shoot apical meristem maintenance, its role in controlling formative divisions in the root seems to be more general and can be observed in both the Col-0 and Ler ecotypes.

## AGO10 is required for HD-ZIP III function in the root vasculature

AGO10 has been suggested to preferentially bind miR165/6 in the SAM and compete with AGO1 for these miRNAs due to a higher affinity (Zhu et al, 2011), thus restricting AGO1-mediated degradation of transcripts of HD-ZIP III TFs, the targets of miR165/6. In their accompanying study, Mirlohi et al (2024) have now established that a similar AGO1-vs-AGO10 competition does occur in the root tip where the promoters of the *MIR165/6* genes are exclusively active in the endodermis (Carlsbecker et al, 2010; Miyashima et al, 2011), whereas AGO10 shows a complementary expression pattern in the stele (Mirlohi et al, 2024, Iyer-Pascuzzi et al, 2011, Fig. 4B). This endodermis- specific expression of miRNA165/6 together with their non-cell autonomous activity (Skopelitis et al, 2018; Vatén et al, 2011) were suggested to establish a miRNA gradient with peak abundance in the endodermis and periphery of the stele, and a minimum in and around the metaxylem cells. Accordingly, HD-ZIP III protein accumulation, with the possible exception of REV, is mostly confined to the inner vascular cells (Carlsbecker et al, 2010). Consistent with the

proposed role of AGO10 in protecting HD-ZIP III transcripts against AGO1-mediated silencing, quantitative real-time PCR analysis demonstrated a decrease in the abundance of HD-ZIP III transcripts in *sgo1* (Fig. EV2A). Moreover, a *phb phv cna* mutant (Prigge et al, 2005) showed root phenotypes remarkably similar to *ago10* mutants (Fig. 4C–G): supernumerary and ectopic differentiated xylem cells as well as increased meristematic stele cell number essentially phenocopied *ago10* mutants, again without compromising root growth (Fig. EV2B), supporting the notion that AGO10 indeed acts through controlling HD-ZIP IIIs. Accordingly, sequestration of miR165/6 by expression of a target mimic (*35S:STTM165/6*) (Yan et al, 2012) alleviated the *sgo1* phenotype (Fig. EV2C). The expression pattern of the pMIR165a:GFP reporter, as well as expression levels of the MIR166b pri-MiRNA did not differ strongly from what was observed in the wild type (Fig. EV2D,E). Taken together, our results suggest that in the absence of AGO10, the miR165/6 gradient could be too shallow to allow for sufficient HD-ZIP III expression, consistent with a dual function of AGO10 in not only sequestering miR165/6 but also promoting their degradation. This latter interpretation is supported by the findings of Mirlohi et al (2024), demonstrating that

*sdn1 sdn2* double mutant roots attenuated *ago10-1*-like xylem defects and that physiological fluctuations of AGO10 and miR165/6 levels are inversely correlated. Importantly, they also demonstrate that HD-ZIP III protein levels are strongly reduced in the absence of AGO10 (Mirlohi et al, 2024).

## AGO10 and HD-ZIP III proteins buffer vascular cytokinin responses

It has been shown that CK is essential for formative divisions in the root vasculature (Mähönen et al, 2000; Mähönen, 2006; De Rybel et al, 2014; Ohashi-Ito et al, 2014). We therefore investigated whether the CK response might be affected in *ago10* mutants using a CK response reporter (Zürcher et al, 2013). Indeed, *pTCSn:2xVenus-NLS* showed an enlarged expression domain in the *sgo1* mutant compared to the wild type as well as increased reporter fluorescence in the stele. In particular, *pTCSn:2xVenus-NLS*-derived fluorescence was observed throughout the procambial tissue in the root meristem, whereas reporter activity in wild-type roots sharply decreased a few cells above the QC (Fig. 5A). The addition of the synthetic CK 6-benzyl adenine (BAP) increased the intensity of

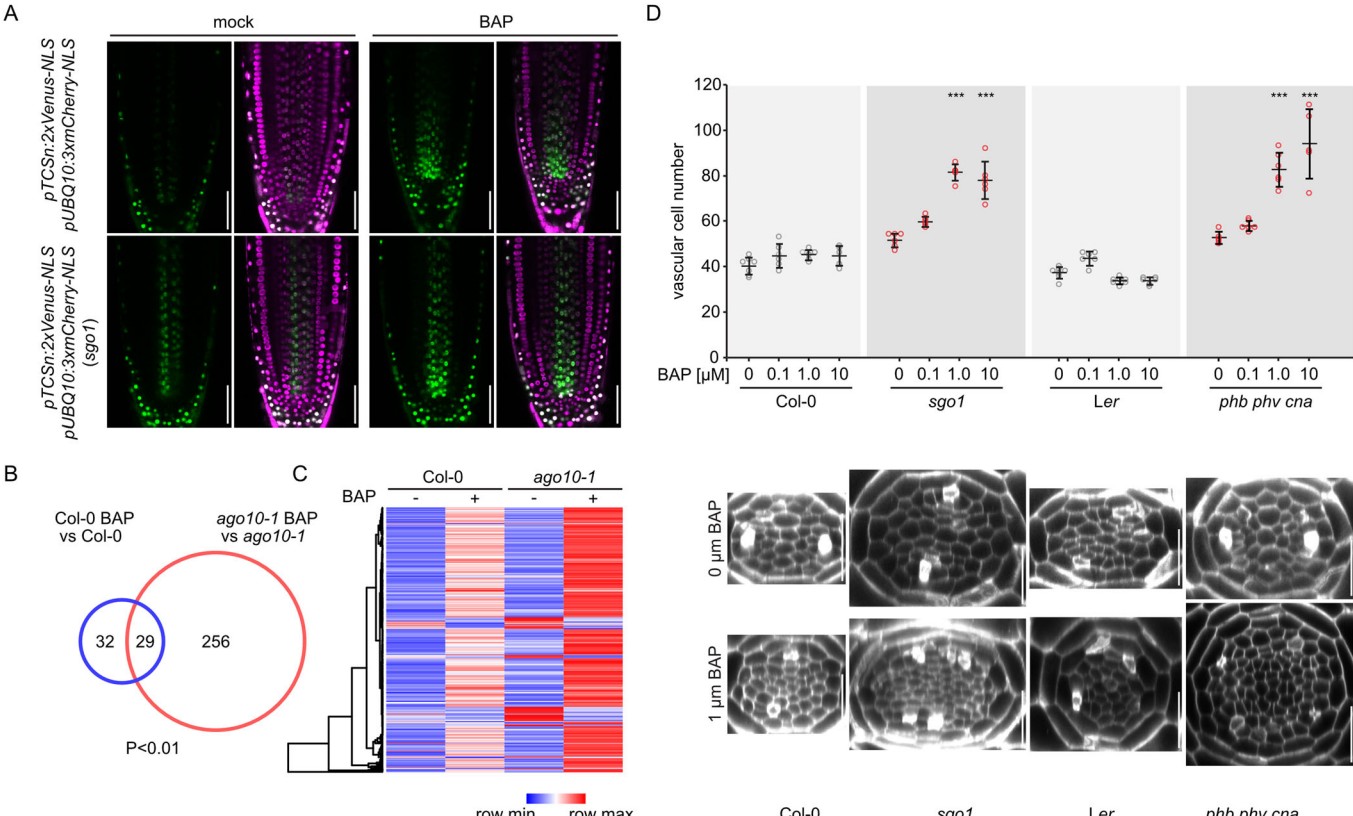

Figure 5. AGO10 and HD-ZIP III proteins buffer vascular cytokinin responses.

(A) Confocal images of *pTCSn:2xVenus-NLS pUB10:3xmCherry-NLS* marker line after mock treatment or after growth on medium supplemented with 0.1 μM BAP (upper panels). Lower panels show the same marker line in the *sgo1* background. (B, C) The *ago10-1* mutants respond more strongly to 2 h of CK treatment based on RNA-seq analysis of root tips (*n* = 3). (B) Venn diagrams show number of genes differentially expressed (*P* < 0.01, see 'Methods' section for details). (C) Heatmap visualization of normalized reads averages from genes in (B) shows increased quantitative response in *ago10-1*. Colour code indicates minimal and maximal value of normalized reads in each row. (D) Ler, and *phb phv cna* roots in response to increasing concentrations of BAP. Data information: graph in (D) depicts means ± s.d. and individual data points. *n* = 6–7 biological replicates. Asterisks indicate statistically significant differences from the respective control (0 μM BAP) within the same genotype based on Tukey's post hoc test after one-way ANOVA (***P* < 0.001). Scale bars in (A), in (C) = 25 μm. Source data are available online for this figure.

reporter-derived fluorescence in both wild type and *sgo1* backgrounds but it did not alter the sharp decrease towards the more mature parts of the wild-type roots (Figs. 5A and EV3A; Movies EV5, EV6). This strongly suggests that the CK response is buffered in the meristem of the root and that this buffering mechanism depends on *AGO10*, and, presumably, on HD-ZIP III TFs. This was corroborated by transcriptomic analysis of the short-term CK response in root tips. After two hours of BAP treatment, 285 genes in *ago10-1* showed differential expression compared to untreated *ago10-1* with a relaxed significance threshold of $P < 0.01$ (see Methods and Dataset EV1). In the Col-0 wild type, only 61 genes showed differential expression after the treatment, approximately half of which were shared with *ago10-1* (Fig. 5B). Moreover, of the genes differentially expressed ($P < 0.01$) in either genotype, almost all responded quantitatively more strongly in *ago10-1* than in Col-0 as shown by clustered comparison of normalized read counts (Fig. 5C). We then tested the response of Col-0, *sgo1*, L*er*, and *phb phv cna* roots to increased CK signalling with respect to formative cell divisions in the root stele. Whereas both the Col-0 and L*er* wild types did not show a significant change in vascular cell number 150 μm above the QC in response to up to 10 μM BAP, *sgo1* and *phb phv cna* showed a dramatic increase of formative cell divisions (Fig. 5D). In extreme cases, we observed roots with more than 100 cells, as opposed to the average of ~40 cells in the corresponding wild-type roots. This result shows that buffering of the promoting effect of CK signalling on formative cell divisions in the stele is mediated by the AGO10-HD-ZIP III module. To assess whether the increased cell division activity in *sgo1* depends on CK signalling, we generated a quadruple mutant carrying lesions in the three root-expressed type B cytokinin response regulators, ARR1, 10, and 12. As previously shown, the *arr1-3 arr10-5 arr12-1* triple mutant (Argyros et al, 2008) shows strongly reduced vascular cell number (Fig. EV3B). The quadruple mutant shows a similar cell file count as *arr1-3 arr10-5 arr12-1*, suggesting that intact CK signalling is required for the *sgo1* phenotype.

## AGO10 controls phenotypic robustness

As an increase in xylem cell number is associated with plant adaption to water stress (Ramachandran et al, 2018), we compared the growth of WT and *sgo1* plants on medium simulating water limitation by addition of polyethylene glycol (PEG) (Verslues et al, 2006). While vascular cell number was reduced under these conditions in both WT and *sgo1* (Appendix Fig. S2A), root growth of *sgo1* plants was more resistant to water-limiting conditions. This was the case when the plants were germinated and grown on PEG-containing medium (Appendix Fig. S2B), as well as after transfer to PEG medium following initial cultivation on control medium (Appendix Fig. S2C). Moreover, adult *sgo1* plants exposed to drought conditions showed strikingly increased survival compared to the Col-0 wild type, although not all plants survived the treatment (Fig. 6A). Likewise, the *phb phv cna* triple mutant showed increased survival in drought experiments, although less pronounced than *sgo1* (Appendix Fig. S2D). Similar to the *phb phv cna* mutant (Fig. EV2B), *sgo1* showed substantially increased root length under the standard growth conditions (half-strength MS with 1% sucrose) compared to the wild type, suggesting that reducing the HD-ZIP III levels is beneficial for growth under lab conditions and water stress. This suggestion is consistent with the

findings, by Mirlohi et al (2024) that AGO10 regulates the global root meristem length/activity.

When we challenged *sgo1* plants with non-optimal carbon supply (0%, 3%, and 4.5% sucrose), these clear growth advantages were no longer apparent, however, and *sgo1* root growth showed a markedly increased variance, particularly under elevated sucrose conditions (Fig. 6B). Both Col-0 and *sgo1* root length was mildly decreased by elevated sucrose compared to the standard medium (1% sucrose), but the coefficient of variance was increased only for *sgo1*. Kernel density estimation revealed two peaks for *sgo1* at 3% sucrose, one approximately at the peak of the 1% sucrose distribution, and one additional peak close to the position of both Col-0 peaks (Fig. 6C). Thus, superior performance of *sgo1* in some conditions appears to be associated with a decrease of phenotypic robustness in others. Similar results were obtained with the *ago10-1* mutant (Fig. EV4). Elevated sucrose is a well-established stress condition for seedlings (Hauser et al, 1995), although the mechanisms by which it causes stress are not known. However, growth on 135 mM sorbitol was not sufficient to elicit similar phenotypes, indicating that the effect exerted by elevated sucrose is not solely osmotic (Fig. EV4). Lugol staining of accumulated starch under different conditions did not reveal any apparent differences between *sgo1* and Col-0 (Appendix Fig. S3). We then assessed whether xylem patterning in Col-0 and *sgo1* was also differentially affected by elevated sucrose availability in the medium. To study potential correlation of the xylem phenotype with growth, we divided *sgo1* roots grown on elevated sucrose conditions in two groups according to root length and separately analysed short and long roots. Whereas Col-0 protoxylem patterning did not change, *sgo1* showed overall increased protoxylem cell numbers when grown in the presence of 4.5% sucrose, again associated with an increased variability. Remarkably, the increase in protoxylem cells and thus increase in variance was largely contributed by the individuals with short root phenotype (Fig. 6D,E), whereas individuals with long roots were indistinguishable from plants grown under control conditions (Fig. 6D,E). The increase in protoxylem cell number of *sgo1* plants grown on elevated sugar levels was associated with a mild, but significant decrease in metaxylem numbers (Fig. EV5A) and thus only a subtle increase in total xylem cell numbers (Fig. EV5B). In contrast to long roots, the short-rooted *sgo1* individuals grown on elevated sugar levels also displayed a more disorganized xylem with an increase in ectopic xylem cells outside of the main xylem axis (Figs. 6F and EV5C). Finally, we assessed vascular cell number in plants grown on 1% and 4.5% sucrose. Interestingly, elevated sugar levels decreased cell number in both WT and *sgo1* roots (Fig. 6G). Again, *sgo1* showed a pronounced loss of phenotypic robustness, and individuals with short roots had markedly fewer cells than those with longer roots. Together, loss of AGO10 function in *sgo1* or *ago10-1* (Fig. EV5D) leads to phenotypic instability under stress conditions. While under control conditions the increase in xylem in *ago10* mutants could at least in part be explained by the increase in formative cell division activity (Fig. 3), this does not apply to the shorter *sgo1* roots under stress conditions, as those plants show the most extreme increase in xylem measured during the course of this study but exhibit vascular cell numbers comparable to those in the wild type under control conditions. Thus, AGO10, presumably by shaping of the miRNA165/6 gradient, is critically required for the control of and the coordination between formative cell division and robust cell fate specification of the vasculature.

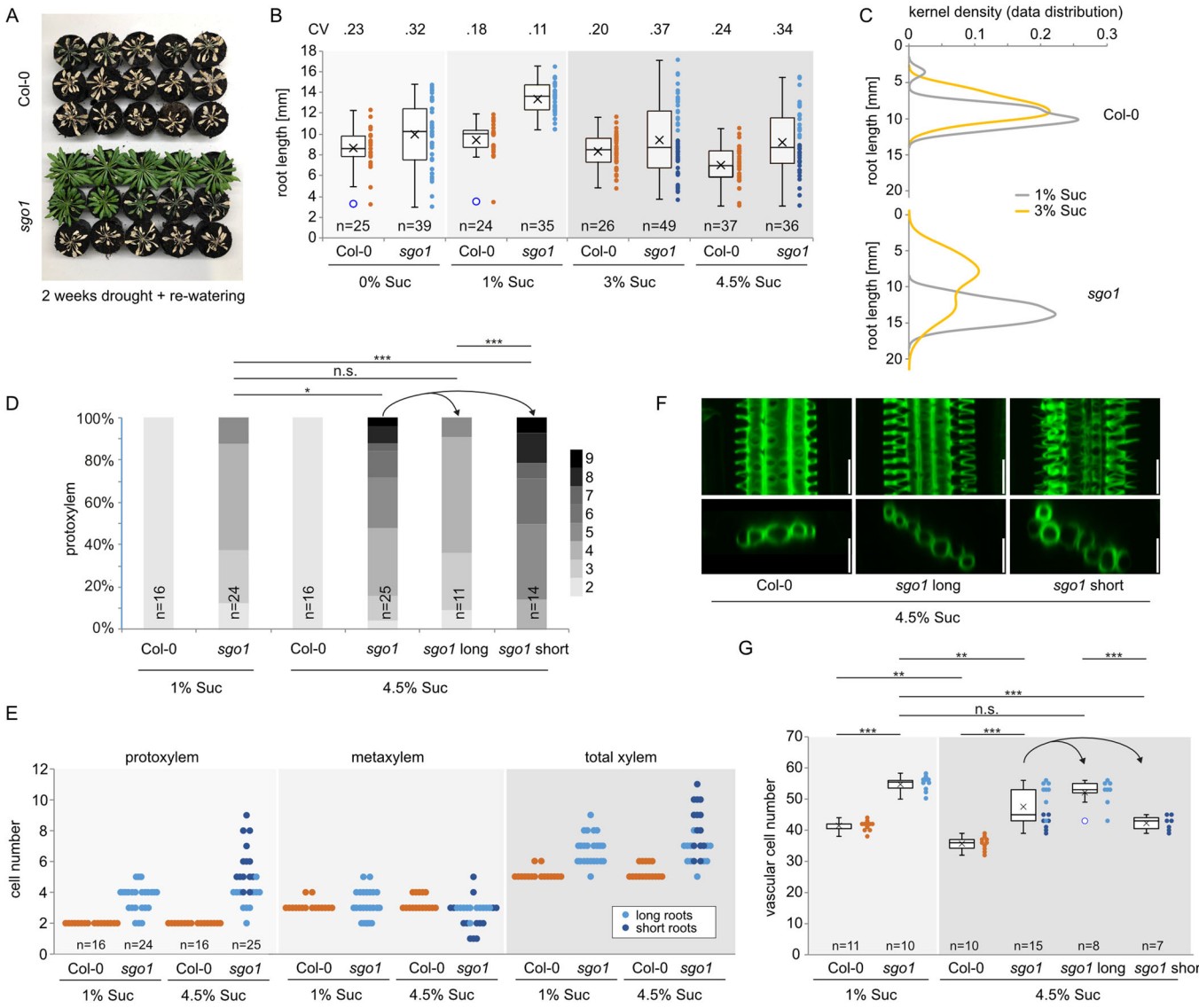

**Figure 6. AGO10 is required for phenotypic robustness.**

(A) *sgo1* mutants show increased survival in drought conditions. Watering of four-week-old plants was stopped for two weeks and then recommenced. Images were taken 10 days after initial re-watering. Plants were grown randomly dispersed in the same tray. (B) Root length of *sgo1* is markedly and uniformly increased under optimal growth conditions but shows increased variance under non-optimal sucrose concentrations. Box plots depict median and upper and lower quartile (boxes), average (cross) and data range (whiskers) of root length under the indicated conditions, dots represent individual data points. CV = coefficient of variance. (C) Kernel density distribution of WT and *sgo1* root length under 1% and 3% sucrose growth regimes. (D) Frequency of roots with the indicated number of protoxylem cells of Col-0 and *sgo1* grown on 1% or 4.5% sucrose. Short and long *sgo1* roots on 4.5% sucrose are depicted separately and combined. Asterisks indicate statistically significant difference based on Mann–Whitney U test (***$P < 0.001$, *$P < 0.05$, n.s. = not significant). Corresponding metaxylem and total xylem graphs are shown in Fig. EV5A, B, respectively.
(E) Alternative display of data from (D) to visualize increase of variance of *sgo1* roots grown under elevated sugar levels. Darker blue dots correspond to data points from short-rooted individuals. (F) Exemplary images depicting xylem phenotypes of WT, long, and short-*sgo1* plants grown under 4.5% sucrose. (G) Quantification of vascular cell files of Col-0 and *sgo1* grown on medium containing the indicated sucrose concentration. Data information: box plots depict median and upper and lower quartile (boxes), average (cross) and data range (whiskers) of root length under the indicated conditions, dots represent individual data points. Darker blue dots correspond to data points from short-rooted individuals. Asterisks indicate statistically significant difference based on Mann–Whitney U test (***$P < 0.001$, **$P < 0.01$, *$P < 0.05$, n.s. = not significant). In each panel, the number of biological replicates is indicated. Source data are available online for this figure.

# Discussion

Plants rely on integration of oriented cell division with developmental patterning of cell identities for the generation of functional organs and tissues. Correct cell fate determination is particularly important in the vasculature to ensure connectivity of the conductive tissues that transport water and nutrients throughout the plant body. In this study, we demonstrate that AGO10 is required for the control of formative cell divisions in the root vasculature as an essential component of miR165/6-mediated patterning. Our results showing that *ago10* mutants are indistinguishable from higher order *hd-zip III* mutants would suggest

that, in the absence of AGO10, miRNA165/6 can spread further from the endodermal source into the stele, curtailing HD-ZIP III expression, a notion also supported by the results of (Mirlohi et al, 2024) in their accompanying study. As previously suggested for the SAM (Yu et al, 2017), Mirlohi et al suggest that AGO10 promotes degradation of miRNAs through SMALL RNA DEGRADING NUCLEASE1 and 2 (SDN1 and SDN2). Consistent with this, miR165/6 and AGO10 levels are anti-correlated (Mirlohi et al, 2024). In addition, miR165/6 levels are increased in the absence of AGO10 (Liu et al, 2009), suggesting that to maintain an instructive miRNA gradient, AGO10 does not only sequester miR165/6 but also triggers their degradation. This would also suggest that, if unchecked by AGO10, miR165/6 might be capable of moving further than previously assumed (Benkovics and Timmermans, 2014; de Felippes et al, 2011). Plant miRNA gradients have been likened to animal morphogens because both allow the dose-dependent, threshold-based cell fate decision and directly act on their targets. A key feature of morphogen gradients is that they act in a scalable manner and allow patterning independent of cell division activity and the cellular layout. Here, we reveal that AGO10 is an essential regulator of root vascular patterning, possibly through expanding the amplitude of the miR165/6 gradient as described in the accompanying study by Mirlohi et al. It will be interesting to see whether comparable "expander" components are general features of gradient decoding and scalability.

Gene regulation by small RNAs gradients is also thought to accomplish canalization of development by reduction of gene expression noise (Schmiedel et al, 2015) and sharpening of expression boundaries (Levine et al, 2007). However, our results demonstrate that unchecked activity of miRNAs can also decrease phenotypic robustness, implying that miRNA gradients have to be finely tuned. Interestingly, a recent study has revealed that miRNA activity can also lead to randomization of cell fate determination, which might serve as a bet-hedging strategy under non-favourable conditions (Plavskin et al, 2016). Futures work should address the ecological and evolutionary consequences of such adjustable plasticity conferred by the AGO10-miR165/6-HD-ZIP III module.

A key remaining question is how the miR165/6-AGO10-HD-ZIP III module controls formative cell divisions. A recent publication has revealed that HD-ZIP IIIs antagonize DOF transcription factors, which promote formative divisions, by repressing *DOF* transcription and cell to cell movement (Miyashima et al, 2019). In addition to these direct effects, our results demonstrating that HD-ZIP IIIs buffer CK responses suggest also an indirect level of control, as expression of the PEAR class of DOF TFs is induced by CK signalling (Miyashima et al, 2019; Smet et al, 2019) and DOF promoters can be targets of ARRs (Smet et al, 2019; Xie et al, 2018; Zubo et al, 2017). Thus, increase in PEAR expression due to loss of HD-ZIP IIIs might at least partially explain the increase in formative cell division in *sgo1* and *phb phv cna*. Interestingly, PHB has been shown to suppress B-type ARR activity (Sebastian et al, 2015), although it should be noted that the interactions between CKs and HD-ZIP III are complex and presumably context-dependent (Sebastian et al, 2015; Dello Ioio et al, 2012). Both pathways also converge on xylem patterning, as CK application suppresses protoxylem specification and CK mutants show protoxylem in place of metaxylem (Argyros et al, 2008; Yokoyama et al, 2006; Mähönen et al, 2000; Mähönen, 2006),

whereas HD-ZIP IIIs gain of function alleles show metaxylem in place of protoxylem (Carlsbecker et al, 2010). In summary, we demonstrate that AGO10 is required for controlling lateral growth and cell fate determination in the root vascular tissue. Our results indicate that AGO10 is an essential component of the miR165/6-HD-ZIP III module, enabling the establishment of an instructive miRNA gradient. We speculate that this intricate non-cell autonomous regulatory circuitry simultaneously provides robustness and flexibility to adapt vascular patterning to both developmental and environmental cues. Notably, HD-ZIP III and AGO10 expression has been shown to respond to water-limiting conditions, favouring increased xylem differentiation (Bloch et al, 2019). A second, presumably vasculature-unrelated function of AGO10 is the control of root meristem length/activity, as shown by Mirlohi et al in the accompanying manuscript. How much of this second function contributes to the phenotypes described here remains to be determined.

## Methods

### Plant material and growth conditions

Plant material used in this study is described in Appendix Table S1. Seeds were surface-sterilized with 1.2% NaOCl in 70% ethanol for five minutes, followed by two rinses in absolute ethanol and air drying under the sterile bench. If not indicated otherwise, plants were grown on half-strength MS medium supplemented with 1% sucrose and 0.9% phytoagar. If appropriate, 6-benzyl adenine was added after autoclaving. For growth under simulated water deficit, half-strength MS plates were infused with PEG as described previously (Verslues and Bray, 2006). Briefly, 25 ml of half-strength MS medium containing 1.5% plant agar were poured into square 12 cm petri dishes and allowed to solidify. This was overlaid with 37.5 ml of liquid half-strength MS medium containing 0, 250, or 400 g/L of PEG 8000 added after autoclaving. Plates were allowed to dry for three days and excess solution was poured off. On plates overlaid with 400 g/L PEG 8000, seeds failed to germinate. For transfer experiments, seedlings were grown on half-strength MS medium without sucrose and transferred to PEG plates after five days of growth. The *ago1-10* mutant was described in Takeda et al, 2008 (Takeda et al, 2008), and the *pAGO10::GFP:AGO10* reference line (also designated *A10^{NX}*) was characterized by Mirlohi et al (2024), in the accompanying study.

### Generation of transgenic plants

For the *pTCSn:2xmVenus-NLS pUBQ10:3xmCherry-NLS* line we first created transgenic plants harbouring construct pCW066 (pUBQ10:3xmCherry-NLS) that were then transformed with construct pCW178 (pTCSn:2xVenus-NLS). Constructs were generated using GreenGate cloning as previously described (Lampropoulos et al, 2013). Modules and oligo sequences can be found in the Appendix Tables S2, S3, respectively.

### Identification of *sgo1*

To identify the mutation causal for the sgo1 phenotype we combined bulked segregant analysis with next-generation

sequencing. We first created a mapping population by crossing *sgo1* to the L*er* wild type. We scored the xylem phenotype of 200 F2 plants by light microscopy, pooled all the plants with *sgo1* phenotype and analysed with a set of PCR-based markers (~5 per chromosome) for Col-0/Ler polymorphisms. We found a clear bias for Col-0 on the bottom of chromosome 5. We then collected 50 plants each with WT and *sgo1* phenotype from an F2 population resulting from a backcross of *sgo1* to Col-0, prepared genomic DNA and sequenced the two pooled samples together with a *sgo1* sample.

Genomic DNA for whole genome sequencing was prepared using CTAB. The samples were ground under liquid nitrogen using pistil and mortar and 70–100 mg plant material was resuspended in 600 μL CTAB buffer (2% CTAB, 1% PVP 4000, 1.4 M NaCl, 100 mM Tris-HCl, pH 8, 20 mM EDTA, pH 8). After a one-hour incubation at 65 °C, samples were cooled down to room temperature and 1 μL RNaseA (1 mg/mL) was added. Samples were incubated for 1 h at 37 °C. 60 μL CHCl₃ were added, gently mixed by inverting the tubes and centrifuged at $5000 \times g$ for 10 min at RT. The polar phase (~500 μL) were transferred to a new reaction tube, and nucleic acids were precipitated by adding 2.5 volumes of 100% ethanol (−20 °C) and incubation for 30 min at −20 °C. Afterwards, precipitated DNA was pelleted at $11,000 \times g$ for 10 min at 4 °C. The supernatant was discarded and 500 μL of 70% ethanol (−20 °C) were added to wash the precipitated DNA, followed by centrifugation at $11,000 \times g$ for 10 min at 4 °C. The supernatant was discarded and the DNA pellet was dried at 55 °C until residual ethanol had evaporated. The DNA pellet was resuspended in 50 μL H2O. Raw sequencing reads were processed with Trimgalore for adaptor removal and quality filtering (https://www.bioinformatics.babraham.ac.uk/projects/trim_galore/ (Martin, 2011); and aligned to the TAIR10 reference genome using bwa (Li and Durbin, 2009). After BAM conversion and indexing with SAMtools (Li et al, 2009), SNPs in the *sgo1* alignment were called using bcftools (Li et al, 2009). The three alignments (*sgo1*, *sgo1*xCol-0 BCF2 with wild-type phenotype, *sgo1*xCol-0 BCF2 *sgo1* phenotype) were then compared and candidate SNPs identified in the region of interest on chromosome five determined by bulked segregant analysis. Only the SNPs in *AGO10/ZWILLE* co-segregated with the *sgo1* phenotype.

## Transcriptome profiling after CK treatment

Col-0 and *ago10-1* seeds were germinated on 1% sucrose, ½ MS plates. Six days after germination, seedlings were sprayed with 1 μM BAP (or left untreated) and after 2 h roots were cut 1–3 mm above root tip and immediately transferred to liquid Nitrogen. RNA extraction was performed using GeneMATRIX Universal RNA/miRNA Purification Kit (Cat. No. E3599 Distributor Roboklon GmbH).

Reads were aligned with RNA STAR (Dobin et al, 2013), read counts were tallied with featureCounts (Liao et al, 2014) from the BAM files using TAIR10.42.gtf as gene annotation file. Differential gene expression was assessed by DESeq2 (Love et al, 2014). The heatmap was visualized with average normalized counts (logCPM) that were hierarchically clustered by complete linkage and Euclidean distance using heatmap.2 in gplots. Data have been deposited at GEO (https://www.ncbi.nlm.nih.gov/geo/) as GSE218342, reviewer access token: wtuzwsyyxpcnzmx.

## Expression analysis by QRT-PCR

Seeds were surface-sterilized with chloride gas, sown on nylon mesh placed on half-strength MS plates containing 0.8% Phyto agar, vernalised for 2 days at 4 °C in the dark and vertically grown under long-day conditions (16 h day/8 h night at 22 °C). About 5 cm long roots were cut by blade from 7-day-old seedlings, transferred to protoplast isolation solution (20 mM MES, 400 mM Mannitol, 20 mM KCl, 1.5% Cellulase, 0.4% Macerozyme, 10 mM CaCl₂) for 15 min to release the root tip from the more mature part of the root at the transition to the elongation zone. Total RNA including miRNA were isolated from root tips with the miRNeasy Mini Kit (QIAGEN, Cat No. 217004) following the standard protocol. Transcript estimation was performed with by quantitative reverse-transcription polymerase chain reaction (qRT-PCR). RNA extractions of three biological replicates per genotype were DnaseI (Invitrogen)-treated to eliminate any residual DNA contamination. A total of 1 μg RNA per sample was reverse transcribed to cDNA according to the manufacturer's instructions (AMV Reverse Transcriptase Native, EURX) using an oligodT primer. QRT-PCRs were run in a Rotor-Gene Q 2plex (Qiagen) cycler on diluted cDNA with four technical replicates and quantified in reference to ACT2. Sybr Green I nucleic acid gel stain (Sigma-Aldrich) was used for detection. Primer efficiency was determined from cDNA serial dilution series. Reactions were amplified at 95 °C initial denaturation for 6 min, then looped through 95 °C for 30 s, 59 °C and 72 °C for 30 s or 10 s for miRNA166 for 40 cycles. Melt curves were obtained from 55 to 95 °C incrementing 1 °C per step. The data were analysed with the 75 Rotor-Gene Q 2plex software and evaluated according to Muller et al (Muller et al, 2002). Statistics were performed according to Rieu and Powers (Rieu and Powers, 2009).

## Genotyping

The presence of the *sgo1* SNP in AGO10 was assessed with a derived cleaved amplified polymporphic sequence (dCAPS) marker (see Appendix Table S2 for oligo sequences). Mutant, but not wild-type amplicons can be cleaved with HindIII, resulting in fragments of 137 bp and 31 bp that were resolved on a 3% agarose gel. Genomic DNA for genotyping was extracted by grinding ~100 mg of leaf tissue in a homogenizer (Retsch mill, QIAGEN) for 30 s and 30 rpm. After addition of 250 μL of gDNA extraction buffer (150 mM Tris-HCl (pH 8), 250 mM NaCl, 25 mM EDTA 0.5% (w/v) SDS), samples were mixed and centrifuged for 15 min at $13,000 \times g$ at room temperature. 150 μL of the supernatant were transferred to a 1.5 mL reaction tube and mixed with 150 μL isopropanol. Precipitated DNA was pelleted by centrifugation at $13,000 \times g$ for 15 min at room temperature. The DNA pellet was washed with 500 μL of 70% ethanol and centrifuged for 5 min as above. The supernatant and residual ethanol was removed and the pellet was air-dried for 5 min before being dissolved in 40 μl of TlowE buffer (10 mM Tris-HCl (pH 8), 0.1 mM EDTA).

## Microscopy

For high-resolution confocal stacks, six-day-old seedlings were fixed in a solution containing paraformaldehyde and SCRI Renaissance 2200 (Renaissance Chemicals, North Duffield, UK)

as described (Musielak et al, 2015). After washing in PBS twice, seedlings were transferred to ClearSee (Ursache et al, 2018; Kurihara et al, 2015) and additionally stained with 0.2% basic fuchsin directly in ClearSee if differentiated xylem was analysed. ClearSee solution was exchanged after a few days. For imaging, samples were in ClearSee in microscopy chambers and stacks of meristems or mature vasculature tissue were acquired with a Leica SP8 with ×63 glycerol objective. SCRI Renaissance 2200 fluorescence was excited with the 405 nm laser line, GFP with 488 nm, YFP with 514 nm, emission was collected between 425 and 475 nm (SCRI Renaissance 2200), 520 and 550 nm (GFP) and 530 and 580 nm (YFP). Stacks were processed with the resliced tool in Fiji and segmented at the indicated positions using CellSeT (Pound et al, 2012).

For pTCSn reporter imaging, longitudinal sections of six-day-old seedlings root meristems were acquired with a Leica SP5 microscope and a ×63 water objective. Venus fluorescence was recorded as described for YFP above, mCherry fluorescence was excited with the 561 nm laser and collected between 575 and 650 nm.

### EdU staining

Plate-grown seedlings were transferred to liquid half-strength MS medium supplemented with 10 µM EdU for the indicated times after which seedlings were either washed in liquid half-strength MS twice and incubated in fresh liquid half-strength MS without EdU or directly fixed in 4% para-Formaldehyde, 0.1% Triton X-100 in 1x PBS for 1 h (Yin and Tsukaya, 2016). After three washes in PBS, the click chemistry reaction was performed with the base-click EdU-Click 488 kit according to the manufacturer's instructions. The seedlings were washed in PBS three times and transferred to ClearSee solution. Cell walls were stained with Calcofluor white (fluorescent Brightener 29, Sigma-Aldrich) in ClearSee according to Ursache et al (Ursache et al, 2018). Imaging of Alexa488-labelled nuclei was performed as described above for GFP, imaging of Calcofluor white-derived cell wall fluorescence as described above for SCRI Renaissance 2200.

### Western blotting

The inflorescence meristems of 40 plants per genotype (Col-0, *sgo1* and *ago10-1*) were ground to a fine powder under liquid nitrogen using a pre-cooled mortar and pestle. For the protein extraction, 150 mg of plant powder was weighed into a pre-cooled 1.5 ml Eppendorf tube and 600 µl of protein extraction buffer (0.7 M sucrose, 0.5 M Tris-HCl, pH 8.0, 5 mM NaEDTA, 0.1 M NaCl, 2& beta-mercaptoethanol, protease inhibitor cocktail) were added. After brief vortexing, the solution was incubated on a rotation wheel for 5 min at 4 °C. After spinning down the plant debris (10 min, 13,000 rpm, 4 °C) the supernatant (~400 µl) was transferred to a fresh tube. To precipitate the proteins one volume Chloroform, four volumes methanol and three volumes of ddH$_2$O were added. After centrifugation (10 min, 10,000 rpm, 15 °C) the upper phase was removed, and 4 volumes Methanol were added. The precipitated proteins were then pelleted by centrifugation (15 min, 13,000 rpm, 15 °C) and as much liquid as possible was removed with a pipette before air drying the protein pellet. 75 µl of resuspension buffer (362.3 mM Tris-Hcl pH 8.0, 10% Glycerol, 3%

SDS) were added, and the samples were incubated at 55 °C and strong shaking for 20 min to dissolve the pellet. The samples were resolved on a 7% SDS-PAGE and then blotted onto a PVDF membrane. The blot was then washed in TBS + 0.1% Tween buffer (TBS-T) for 5 min and blocked for 30 min with 5% milk powder dissolved in TBS-T. Afterwards, the membrane was washed thrice with TBS-T for 5 min before the primary anti-AGO10 antibody (Iki et al, 2018) (1:10,000 in 3% BSA dissolved in TBS-T) was added for overnight incubation at 4 °C. Then, the membrane was washed as before and was incubated with the secondary anti-rabbit-HRP antibody (Pierce, 1:5000 in 5% milk powder dissolved in TBS-T) for 1 h at room temperature. After washing the membrane as before, detection was carried out using the SuperSignal West Dura Extended Duration Substrate (Thermo Fisher). Chemiluminescence was detected using an Intas imaging system.

### Lugol staining experiment

Surface sterilized Col-0 and *sgo1* seeds were grown on ½ MS standard plates (1% Sucrose) and plates supplemented with 4.5% Sucrose in long-day conditions (16 h light/8 h dark at 22 °C). After 6 h in light or prolonged darkness (10 h), 6-day-old seedlings were incubated in Lugol solution (Sigma-Aldrich) for 30 s. The roots were then mounted in Clearing Solution (4:3:1 Chloralhydrate:-H$_2$O:Glycerol) and imaged using a Zeiss Imager M2 epifluorescence microscope (×20 objective, DIC). We quantified the number of cell files containing starch granules stained by Lugol solution in the root cap. If it contained stained starch granules, the detaching outermost layer of the root cap was counted as well, as we could not rule out its detachment due to sample handling.

## Data availability

The datasets produced in this study are available at GEO as GSE218342.

## Peer review information

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

## Acknowledgements

The authors would like to thank Dolf Weijers, Bert De Rybel, Ari Pekka Mähönen, Olivier Voinnet, and Thomas Greb for sharing materials. Furthermore, the authors are thankful to Ingrid Lohmann for providing microscope access, and to Rosa Lozano-Durán for critical reading of the manuscript and discussions. Research was supported by the German Research Foundation (DFG) with grants WO 1660/2 and WO 1660/6 to SW.

## Author contributions

**Nabila El Arbi**: Formal analysis; Investigation; Visualization; Methodology. **Ann-Kathrin Schürholz**: Formal analysis; Investigation; Visualization; Methodology. **Marlene U Handl**: Formal analysis; Investigation; Visualization; Methodology. **Alexei Schiffner**: Investigation; Methodology. **Inés Hidalgo Prados**: Investigation; Methodology. **Liese Schnurbusch**: Formal analysis; Investigation; Visualization; Methodology. **Christian Wenzl**: Investigation; Methodology. **Xin'Ai Zhao**: Resources; Investigation; Methodology. **Jian Zeng**: Resources; Investigation; Methodology. **Jan U Lohmann**: Conceptualization; Resources; Supervision; Methodology. **Sebastian Wolf**: Conceptualization; Data curation;

Formal analysis; Supervision; Funding acquisition; Investigation; Visualization; Writing—original draft; Project administration; Writing—review and editing.

## Disclosure and competing interests statement

The authors declare no competing interests.

# Expanded View Figures

**Figure EV1.   Related to Fig. 4. *SGO1* encodes AGO10, which is essential for root vascular patterning in Col-0 and L*er*.**

(**A**) Basic fuchsin staining of lignified xylem cells in L*er* and the AGO10 mutant *zll-3*. Upper panels are xy projections of a confocal stack, lower panels optical xy sections through the same stack. Asterisks denote cells with protoxylem differentiation, arrows point to ectopic xylem strands. Scale bar = 25 μm. (**B–D**) Frequency of roots with the indicated number of protoxylem (**B**), metaxylem (**C**) or total xylem (**D**) cells in L*er* and zll-3. Asterisks indicate statistically significant difference from Col-0 based on Mann–Whitney U test (**$P < 0.01$, *$P < 0.05$). (**C**) Quantification of vascular cell number in cross section of confocal stacks at 15 μm, 22 μm, and 150 μm distance from the quiescent centre (QC) cells in L*er* and *zll-3* meristems. Graph depicts means ± s.d. and individual data ($n = 10$–12). Letters in graph indicate statistically significant differences based on Tukey's post hoc test after one-way ANOVA. (**D**) Frequency of roots with the indicated protoxylem cell number in L*er*, *sgo1* and F1 plants of the indicated crosses, demonstrating that *zll-3* and *sgo1* are allelic to each other. Asterisks indicate statistically significant differences from L*er* based on Dunn's post hoc test with Benjamini–Hochberg correction after Kruskal–Wallis modified U test (**$P < 0.01$). (**E**) Immunodetection of AGO10 in Col-0, *sgo1*, and *ago10-1*. Lower panel depicts the Ponceau-stained membrane for loading control.

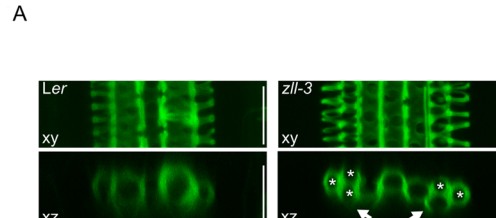

A

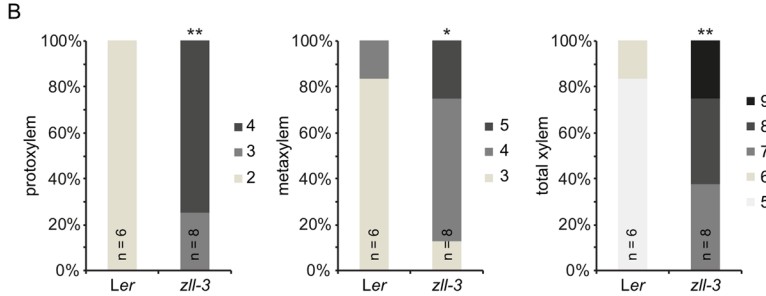

B

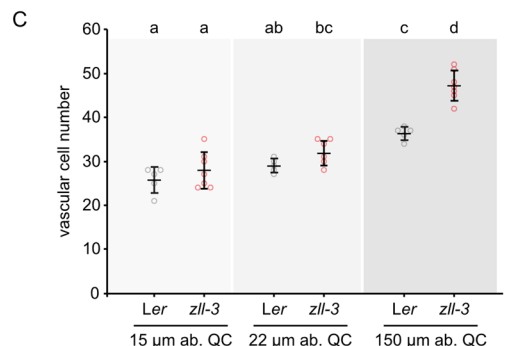

C

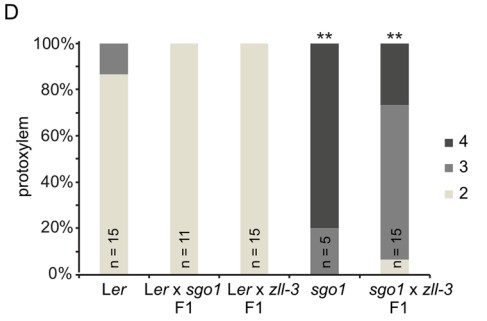

D

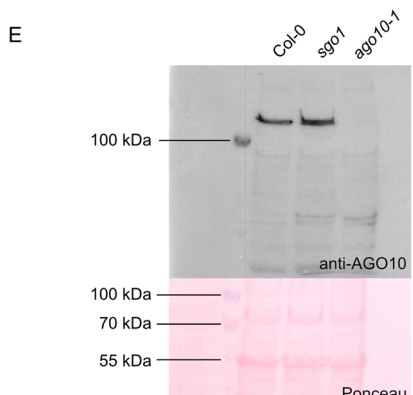

E

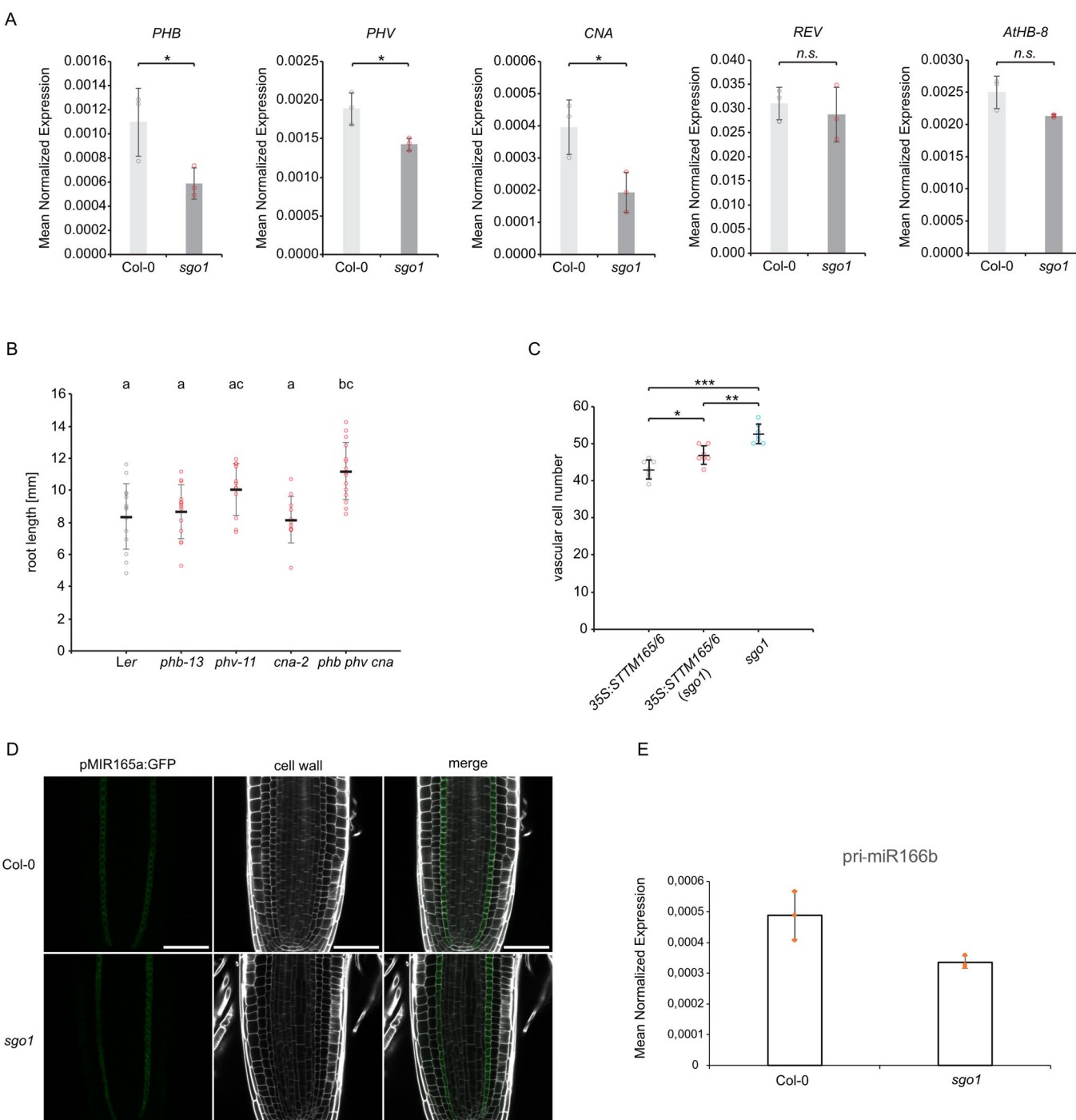

**Figure EV2. Related to Fig. 4. AGO10 is required for HD-ZIP III-mediated vascular patterning.**

(A) Quantitative real-time reverse transcribed PCR (pRT-PCR) analysis indicates reduced HD-ZIP III transcript abundance in *sgo1*. Bars indicate average of Mean normalized expression (MNE) values from three independent biological replicates ± s.d., individual data points are indicated. Asterisks denote statistically significant differences based on a two-tailed student's t-test of log2-transformed MNE values according to Rieu and Powers (Rieu and Powers, 2009). (B) Root length quantification of Ler, *phb-11*, *phv-13*, *cna-2*, and *phb phv cna* 7 days after germination. Graph denotes means ± s.d., individual data points are indicated. Letters indicate statistically significant differences based on Tukey's post hoc test after one-way ANOVA. (C) Quantification of vascular cell number in *35S:STTM165/6* expressing a miR165/6 target mimic, *sgo1*, and the target mimic line in the *sgo1* background. Asterisks indicate statistically significant difference based on Tukey's HSD test following one-way ANOVA with (***P < 0.001, **P < 0.01, *P < 0.05). n = 7. (D) Expression pattern of the pMIR165a:GFP reporter is not altered in the *sgo1* background. Scale bars = 50 μm. (E) Levels of the pri-miR165/6 transcripts are not significantly altered in the *sgo1* background. QPCR results with a primer pair binding all pri-miR transcripts are shown. Bars indicate average of Mean normalized expression values from three independent biological replicates ± s.d., individual data points are indicated.

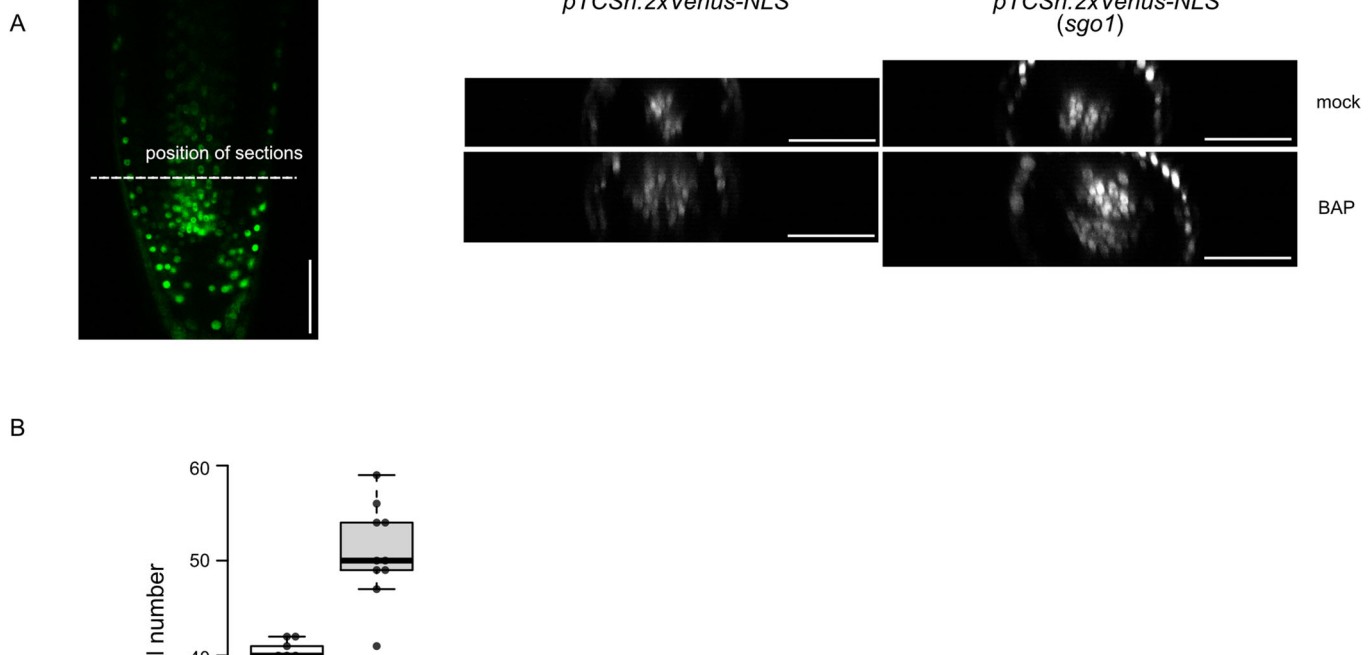

**Figure EV3.   Related to Fig. 5. Increased vascular cell numbers in sgo1 depend on intact cytokinin signalling.**

(**A**) Confocal sections of *pTCSn:2xVenus-NLS pUB10:3xmCherry-NLS* marker line after mock treatment or after growth on medium supplemented with 0.1 μM BAP. Scale bar = 50 μm. (**B**) Lesion of three of the main type-B ARR CK response regulators, ARR1, ARR10, and ARR12 (Argyros et al, 2008), results in a dramatic reduction of vascular cell numbers in both the Col-0 and *sgo1* background, respectively. Box plots depict median and upper and lower quartile (boxes), average (cross) and data range (whiskers), excluding outliers, of root length under the indicated conditions, dots represent individual data points. *n* = 8–10.

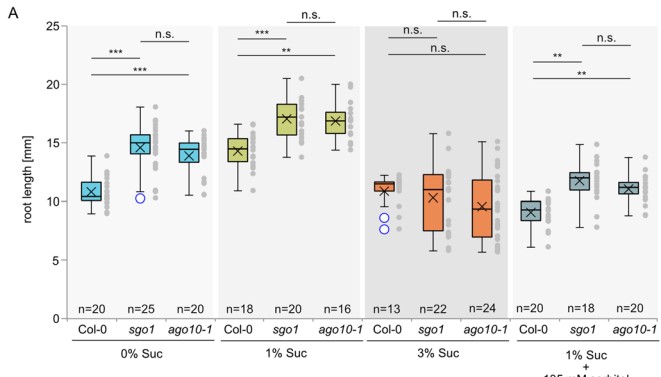

**Figure EV4. Related to Fig. 6. AGO10 is required for phenotypic robustness.**

(A) Root length of Col-0, *sgo1*, and *ago10-1* at 6DAG under the indicated conditions. Box plots depict median and upper and lower quartile (boxes), average (cross) and data range (whiskers) of root length under the indicated conditions, dots represent individual data points. Asterisks indicate statistically significant difference based on Tukey's HSD test following one-way ANOVA with (***$P < 0.001$, **$P < 0.01$, n.s. = not significant). Significance differences are only indicated within a given condition.

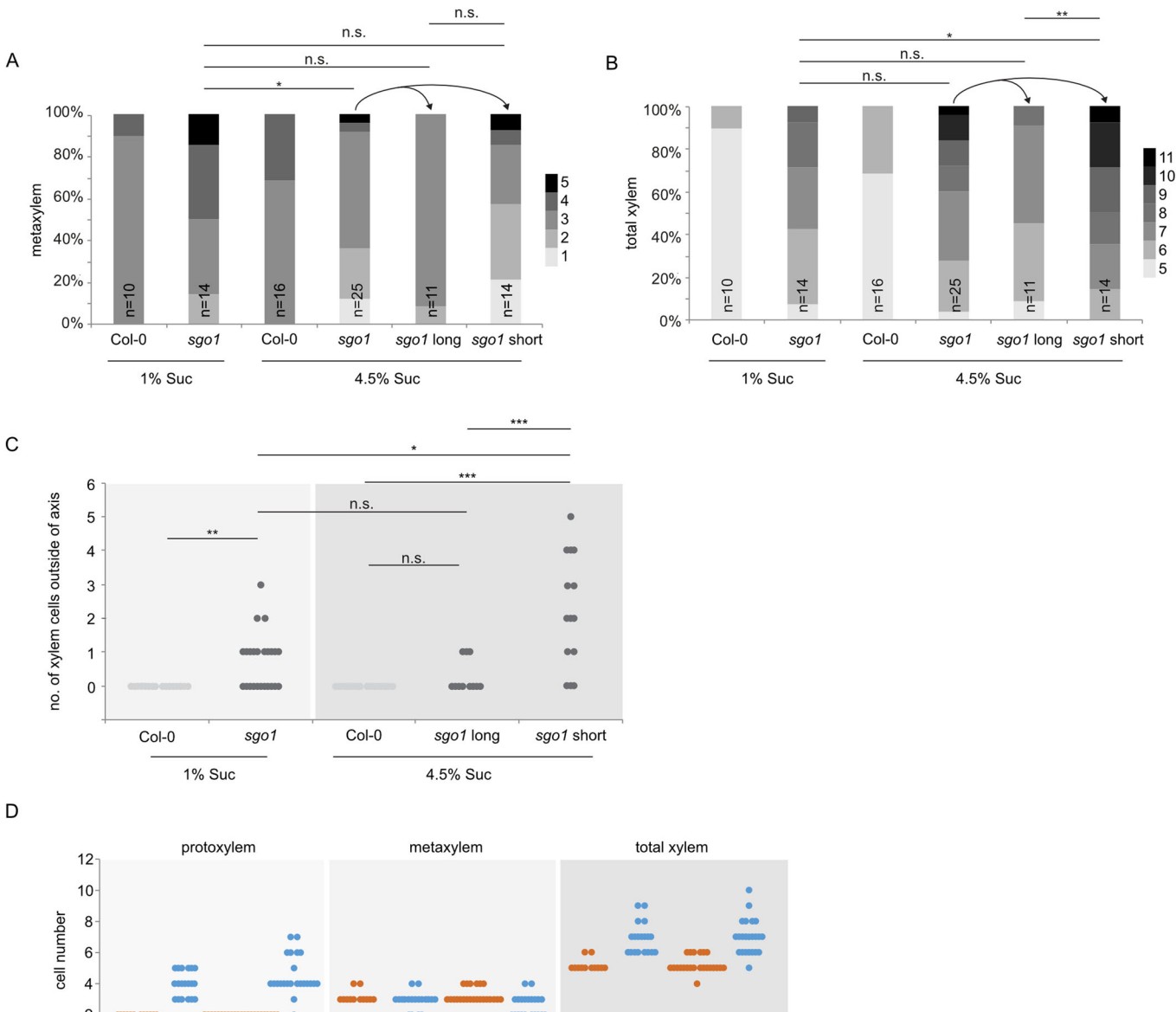

**Figure EV5.  Related to Fig. 6. AGO10 is required for phenotypic robustness.**

(**A, B**) Frequency of roots with the indicated number of metaxylem (**A**) and total xylem (**B**) of Col-0 and *sgo1* grown on 1% or 4.5% sucrose. Short and long *sgo1* roots on 4.5% sucrose are depicted separately and combined. Asterisks indicate statistically significant difference based on Mann–Whitney U test (**$P < 0.01$, *$P < 0.05$, n.s. = not significant). (**C**) Plot of individual roots grown under the indicated conditions according to the number of ectopic xylem cells differentiating outside of the xylem axis, i.e. in procambial position. Asterisks indicate statistically significant differences based on Dunn's post hoc test with Benjamini–Hochberg correction after Kruskal–Wallis modified U test (***$P < 0.001$, **$P < 0.01$, *$P < 0.05$, n.s. = not significant). (**D**) Plot of differentiated xylem cell number of Col-0 and *ago10-1* roots grown under the indicated conditions.

