## [Peer Review File · The EMBO Journal]

ARGONAUTE10 controls cell fate specification and formative cell divisions in the Arabidopsis root

Nabila El Arbi, Ann-Kathrin Schürholz, Marlene Handl, Alexei Schiffner, Inés Hidalgo Prados, Liese Schnurbusch, Christian Wenzl, Xin'Ai Zhao, Jiang Zheng, Jan Lohmann, and Sebastian Wolf

Corresponding author(s): Sebastian Wolf (sebastian.wolf@zmbp.uni-tuebingen.de)

Review Timeline:

Submission Date:	13th Jan 23
Editorial Decision:	21st Mar 23
Revision Received:	15th Dec 23
Editorial Decision:	12th Feb 24
Revision Received:	20th Feb 24
Accepted:	22nd Feb 24

Editor: William Teale

Transaction Report:

Dear Dr. Wolf,

Thank you again for the submission of your manuscript entitled "AGO10 controls cell fate specification and formative cell divisions in the Arabidopsis root" (EMBOJ-2023-113494) and for your patience during the review process. We have now received the reports from the referees, which I copy below.

Based on the overall interest expressed in the reports, I would like to invite you to address the comments of all referees in a revised version of the manuscript. I should add that it is The EMBO Journal policy to allow only a single major round of revision and that it is therefore important to resolve the main concerns at this stage. I believe the concerns of the referees are reasonable and addressable, please contact me if you have any questions, need further input on the referee comments or if you anticipate any problems in addressing any of their points. I am happy to set up a Zoom call to discuss any of this if you think it would be helpful. Please, follow the instructions below when preparing your manuscript for resubmission.

I would also like to point out that as a matter of policy, competing manuscripts published during this period will not be taken into consideration in our assessment of the novelty presented by your study ("scooping" protection). We have extended this 'scooping protection policy' beyond the usual 3 month revision timeline to cover the period required for a full revision to address the essential experimental issues. Please contact me if you see a paper with related content published elsewhere to discuss the appropriate course of action.

Again, please contact me at any time during revision if you need any help or have further questions.

Thank you very much again for the opportunity to consider your work for publication. I look forward to your revision.

Best regards,

William

William Teale, Ph.D.
Editor
The EMBO Journal

When submitting your revised manuscript, please carefully review the instructions below and include the following items:

- 1) a .docx formatted version of the manuscript text (including legends for main figures, EV figures and tables). Please make sure that the changes are highlighted to be clearly visible.
- 2) individual production quality figure files as .eps, .tif, .jpg (one file per figure).
- 3) a .docx formatted letter INCLUDING the reviewers' reports and your detailed point-by-point response to their comments. As part of the EMBO Press transparent editorial process, the point-by-point response is part of the Review Process File (RPF), which will be published alongside your paper.
- 4) a complete author checklist, which you can download from our author guidelines ([https://wol-prod-cdn.literatumonline.com/pb-assets/embo-site/Author Checklist%20-%20EMBO%20J-1561436015657.xlsx](https://wol-prod-cdn.literatumonline.com/pb-assets/embo-site/Author%20Checklist%20-%20EMBO%20J-1561436015657.xlsx)). Please insert information in the checklist that is also reflected in the manuscript. The completed author checklist will also be part of the RPF.
- 5) Please note that all corresponding authors are required to supply an ORCID ID for their name upon submission of a revised manuscript.
- 6) We require a 'Data Availability' section after the Materials and Methods. Before submitting your revision, primary datasets produced in this study need to be deposited in an appropriate public database, and the accession numbers and database listed under 'Data Availability'. Please remember to provide a reviewer password if the datasets are not yet public (see <https://www.embopress.org/page/journal/14602075/authorguide#datadeposition>). If no data deposition in external databases is needed for this paper, please then state in this section: This study includes no data deposited in external repositories. Note that the Data Availability Section is restricted to new primary data that are part of this study.

Note - All links should resolve to a page where the data can be accessed.

8) For data quantification: please specify the name of the statistical test used to generate error bars and P values, the number (n) of independent experiments (specify technical or biological replicates) underlying each data point and the test used to calculate p-values in each figure legend. The figure legends should contain a basic description of n, P and the test applied. Graphs must include a description of the bars and the error bars (s.d., s.e.m.).

9) We would also encourage you to include the source data for figure panels that show essential data. Numerical data can be provided as individual .xls or .csv files (including a tab describing the data). For 'blots' or microscopy, uncropped images should be submitted (using a zip archive or a single pdf per main figure if multiple images need to be supplied for one panel). Additional information on source data and instruction on how to label the files are available at .

10) We replaced Supplementary Information with Expanded View (EV) Figures and Tables that are collapsible/expandable online (see examples in <https://www.embopress.org/doi/10.15252/emboj.201695874>). A maximum of 5 EV Figures can be typeset. EV Figures should be cited as 'Figure EV1, Figure EV2' etc. in the text and their respective legends should be included in the main text after the legends of regular figures.

12) Our journal encourages inclusion of *data citations in the reference list* to directly cite datasets that were re-used and obtained from public databases. Data citations in the article text are distinct from normal bibliographical citations and should directly link to the database records from which the data can be accessed. In the main text, data citations are formatted as follows: "Data ref: Smith et al, 2001" or "Data ref: NCBI Sequence Read Archive PRJNA342805, 2017". In the Reference list, data citations must be labeled with "[DATASET]". A data reference must provide the database name, accession number/identifiers and a resolvable link to the landing page from which the data can be accessed at the end of the reference. Further instructions are available at .

Further instructions for preparing your revised manuscript:

- a point-by-point response to the referees' comments, with a detailed description of the changes made (as a word file).
- a word file of the manuscript text.

- individual production quality figure files (one file per figure)
 - a complete author checklist, which you can download from our author guidelines (<https://www.embopress.org/page/journal/14602075/authorguide>).
 - Expanded View files (replacing Supplementary Information)
- Please see out instructions to authors
<https://www.embopress.org/page/journal/14602075/authorguide#expandedview>

We realize that it is difficult to revise to a specific deadline. In the interest of protecting the conceptual advance provided by the work, we recommend a revision within 3 months (19th Jun 2023). Please discuss the revision progress ahead of this time with the editor if you require more time to complete the revisions. Use the link below to submit your revision:

Referee #1:

In this manuscript El Arbi and colleagues describe identification of a new allele of ARGONAUT10 gene and propose its role in control of root growth and patterning of the vascular tissue. AGO10 has been shown to destabilize MIR165/6 molecules and allow for accumulation of class III HD-ZIP proteins in different developmental contexts. Here, the authors show that the roots of sgo1 show must go on 1 (sgo1) phenotypically mimic combinatorial phb cna phv HD-ZIP mutant (increased number of vascular cell files, increased number of xylem files). This is very interesting and provides new insights to the long-studied regulatory pathway influencing vascular patterning in root. The authors, also, identified a drought resistance phenotype in the sgo1 mutant. However, at this stage it is unclear how described morphological changes contribute to the described drought resistance.

Major comments:

The authors show lower transcript levels for all HD-ZIPs in the sgo1. It would be important to test protein accumulation of PHB and CNA in the vascular tissue in the mutant background. Are protein levels lower throughout the vascular tissue or is the protein accumulation restricted to some smaller domain? Alternatively RNA in-situ analysis could be used to describe changes in HD-ZIP mRNA distribution in the mutant background.

L308-312 The authors suggest that the role of SGO1/AGO10 is to control spatial distribution of miRNA165/6 and therefore class III HD-ZIP proteins. In the past, AGO10 has been shown to interact with other miRNAs, e.g. miR398. Does reduction of PHB, PHV and CNA protein levels explain draught resistance phenotype of sgo1? The authors should directly test hd-zip combinatorial mutant in such conditions (PEG, soil).

The authors should test if the MIR165 expression pattern remains unchanged in the sgo1 mutant background. In figure 5 the panel A the Ck treatment was performed in sgo1 mutant background but then, in panel B, the RNA-seq is performed using ago10-1. This should be clarified. Which is the purpose of the RNA-seq? Can authors provide lists of DEGs and GO analysis? Is there any interesting information regarding CK and vascular development to highlight? What is the reason for the carbon-supply study? I will suggest to the authors to stain starch with lugol, maybe sgo1 mutants have some problem with sugar metabolism. Are genes of calvin cycle deregulated? This together with the miRNA165 expression levels would help to understand better the phenotype.

Optional:

Have the authors considered grafting experiments to understand the role of root and shoot in drought resistance of the sgo1/ago10?

Minor

- The authors suggest that AGO10 is required for the maintenance of miRNA 165/6 gradient. They show in figure 4 the expression localization of miRNA165a and AGO10 protein expression pattern but it is not clear which background are they using. Is Col 0 or ago10? If the reporter lines are in the mutant background, the authors should show the same reporter lines in

the WT background.

- Indicate genetic background of reporter lines in Fig 4B
- L112 Provide a full name for the RLP4, SDN1, SDN2 etc, when it's used for the first time.
- L206 Can the authors indicate if the mutation was dominant or recessive
- L238-242 Shall the authors include the reference to figure 4B showing expression pattern of AGO10 in the stele?
- Higher-order combinatorial class III hd-zip mutants often show a triarch pattern in vascular tissue (three phloem poles). Is such phenotype ever observed in the sgo1?
- References with formatting error: Mahonen, 2006 (line 65, 167, 267,413), Mirlohi et al 2022 (line 158, 241, 369, 371), Ramachandran et al 2020 (line 299)...

Referee #2:

In this manuscript, El Arbi et al. make a very valuable contribution to understanding the role of AGO10 in the control and the coordination of formative cell division and cell fate specification of the root vasculature. Also the manuscript is well written and the experiments are well justified, I have some comments to be addressed:

It would have been great if the figures would have captions under them to quickly recognize which figure is which.

Major comments:

- Line 111. I think it would be great if the authors elaborate a bit more on how the gene was discovered, the appearance is a bit vague in the text.
- Figure 2A. Also, the differences seem to be clear in the optical sections, it would be great to provide some z-stack reconstructions of a few cell layers to avoid the bias of sporadic appearance of EdU incorporation.
- Figure 3E and F. I am not sure if it's easy to follow the "ectopic divisions" here. Please elaborate how these quantifications were done and what arrows are pointing at. At the same time, it would be great to see WT corresponding images and 3D reconstruction of TMO5 expression for a better comparison.
- Figure 5B. I think it would be very informative to elaborate of what groups of genes the authors identify here, perhaps providing some GO terms at least.
- Figure 6A. The response to drought phenotype seems to be impressive in sgo1 mutant. It would be important to discuss about the possible explanations for it in the Discussion section. Is it connected to more xylem formation, enhanced rate of formative divisions?
- Line 366. How do the miRNA165/6 and HD-ZIPIII TFs behave in ago10 background? It would be important to visualize it to support the statement that "in the absence of AGO10, miRNA165/6 can spread further from the endodermal source into the stele, curtailing HD-ZIP III expression."
- Figure 5A. It would be very informative to provide optical sections of TCSn marker in WT and sgo1 mutant to appreciate better the spatial distribution of cytokinin signalling, especially after BAP treatment.

Minor comments:

- Please check the use of the definite articles throughout the text.
- Line 34. ".....acts by buffering cytokinin responses and restricting xylem differentiation."
- Line 36. Too many "show"
- Line 39. "Thus, AGO10 is required for the control of and coordination...."
- Line 63. Introduce the abbreviation for TFs.
- Line 81. Define TFs.
- Line 130. "We quantified the cell number close to the stem cell region." Please correct.
- Line 358. I assume the authors mean here rather xylem cell fate specification, as they don't focus on phloem phenotypes. Also, the divisions are affected in procambium, the authors focus primarily on xylem specification.

Referee #3:

El Arbi et al present a previously underexplored role for the Arabidopsis AGO10 gene in controlling root vascular development and phenotypic robustness. The work is performed at the highest level and the story is very cohesive. The notion that AGO10 controls periclinal/formative divisions, and limits phenotypic variability is interesting and important. I have only minor comments and am generally very supportive of publication.

1. What is unclear from the work is what the significance is of more/less phenotypic plasticity as conferred by different AGO10 levels. As presented, ago10 mutants perform better in drought conditions, and roots have increased vascular cell numbers without compromising root length. Could the authors explore fitness penalties of the ago10 mutation to better understand the role of phenotypic robustness/buffering provided by AGO10?

2. While perhaps clear to specialists, Figure 6C would need more explanation for the uninitiated.

3. While the effect of AGO10 is predicted to be mediated by Class III HD-ZIP factors, the authors do not in fact show if/how these are affected. I would encourage the authors to show the accumulation pattern of at least one of these in the ago10 mutant.

Dear Dr. Wolf,

Thank you again for the submission of your manuscript entitled "AGO10 controls cell fate specification and formative cell divisions in the Arabidopsis root" (EMBOJ-2023-113494) and for your patience during the review process. We have now received the reports from the referees, which I copy below.

Based on the overall interest expressed in the reports, I would like to invite you to address the comments of all referees in a revised version of the manuscript. I should add that it is The EMBO Journal policy to allow only a single major round of revision and that it is therefore important to resolve the main concerns at this stage. I believe the concerns of the referees are reasonable and addressable, please contact me if you have any questions, need further input on the referee comments or if you anticipate any problems in addressing any of their points. I am happy to set up a Zoom call to discuss any of this if you think it would be helpful. Please, follow the instructions below when preparing your manuscript for resubmission.

I would also like to point out that as a matter of policy, competing manuscripts published during this period will not be taken into consideration in our assessment of the novelty presented by your study ("scooping" protection). We have extended this 'scooping protection policy' beyond the usual 3 month revision timeline to cover the period required for a full revision to address the essential experimental issues. Please contact me if you see a paper with related content published elsewhere to discuss the appropriate course of action.

When preparing your letter of response to the referees' comments, please bear in mind that this will form part of the Review Process File, and will therefore be available online to the community. For more details on our Transparent Editorial Process, please visit our website: <https://www.embopress.org/page/journal/14602075/authorguide#transparentprocess> ;

Again, please contact me at any time during revision if you need any help or have further questions.

Thank you very much again for the opportunity to consider your work for publication. I look forward to your revision.

Best regards,

William

Referee #1:

In this manuscript El Arbi and colleagues describe identification of a new allele of ARGONAUT10 gene and propose its role in control of root growth and patterning of the vascular tissue. AGO10 has been shown to destabilize MIR165/6 molecules and allow for accumulation of class III HD-ZIP proteins in different developmental contexts. Here, the authors show that the roots of *sgo1* must go on 1 (*sgo1*) phenotypically mimic combinatorial *phb cna phv* HD-ZIP mutant (increased number of vascular cell files, increased number of xylem files). This is very interesting and provides new insights to the long-studied regulatory pathway influencing vascular patterning in root. The authors, also, identified a drought resistance phenotype in the *sgo1* mutant. However, at this stage it is unclear how described morphological changes contribute to the described drought resistance.

Major comments:

1. The authors show lower transcript levels for all HD-ZIPs in the *sgo1*. It would be important to test protein accumulation of PHB and CNA in the vascular tissue in the mutant background. Are protein levels lower throughout the vascular tissue or is the protein accumulation restricted to some smaller domain? Alternatively RNA in-situ analysis could be used to describe changes in HD-ZIP mRNA distribution in the mutant background.

>>We agree with the reviewer that this is an important point which now has been comprehensively addressed in the accompanying manuscript by Mirlohi et al., demonstrating that PHB-GFP indeed strongly reduced in *ago10* mutants.

We had initiated similar experiments and performed crosses between *ago10-1/sgo1* and a PHB-YPET line that became available. However, we were notified by the Voinnet laboratory that this line seems to show a general HD-ZIP III loss-of-function phenotype presumably due to transgene-dependent co-suppression. We could confirm ectopic xylem that shows exclusively protoxylem-like patterning independent of the presence of AGO10. We therefore stopped these experiments and now refer to the results of Mirlohi et al.

2. L308-312 The authors suggest that the role of SGO1/AGO10 is to control spatial distribution of miRNA165/6 and therefore class III HD-ZIP proteins. In the past, AGO10 has been shown to interact with other miRNAs, e.g. miR398. Does reduction of PHB, PHV and CNA protein levels explain drought resistance phenotype of *sgo1*? The authors should directly test *hd-zip* combinatorial mutant in such conditions (PEG, soil).

>> We have performed these experiments, and the results are included as Appendix Figure S2D. The *phv phb cna* triple mutant (in *er-2* background, Prigge et al, 2005) shows a subtle drought resistance phenotype consistent with our hypothesis that *ago10* phenotypes are mainly related to reduced HD-ZIP III levels. We note that the dose-dependency of the *hd zip* III mutant phenotypes (e.g. Carlsbecker et al., 2010, Supplemental Figure 19) complicates the interpretation of this result, therefore we refrain from discussing it extensively in the manuscript.

The authors should test if the MIR165 expression pattern remains unchanged in the *sgo1* mutant background.

>> We have crossed the transcriptional miRNA reporter line pMIR165a:GFP with *sgo1* and observed that its spatial expression pattern was not affected by the absence of AGO10. In addition, QPCR with

a primer pair directed against the transcripts of the other main root-expressed MIR165/6 member, MIR166b (Carlsbecker et al., 2010) revealed largely unaltered expression levels. These results are now included as Figure EV2D and E.

In figure 5 the panel A the Ck treatment was performed in *sgo1* mutant background but then, in panel B, the RNA-seq is performed using *ago10-1*. This should be clarified. Which is the purpose of the RNA-seq?

>>Our rationale for the RNA-seq experiment was to test cytokinin hypersensitivity of *ago10* mutants. We reason that the quantitative increase in number of significant DEGs as well as their more pronounced response provides quantitative molecular support for the hypothesis that AGO10 buffers CK responses. We therefore also initially refrained from extensively discussing the identity of the DEGs but now provide lists of DEGs and GO analysis as a EV Dataset (see below).

The use of two different *ago10* alleles was largely due to historical and organizational reasons. We note, however that the *sgo1* and *ago10-1* alleles behave identically throughout all experiments in our study and the accompanying study of Mirlohi et al, under the respective growth conditions of each laboratory.

.

Can authors provide lists of DEGs and GO analysis?

>>We have added this information as Dataset S1

Is there any interesting information regarding CK and vascular development to highlight?

>> Most notable is perhaps the consistent upregulation of several JAZ transcripts (and consequently, GO enrichment of JA-related categories) as a role for jasmonic acid signalling in vascular development has been shown recently. However, since upregulation of JAZ is not straightforward to interpret as an up- or downregulation of JA signalling, we refrain from discussing this further in the manuscript and intend to follow this lead with future experiments.

What is the reason for the carbon-supply study?

>>We initially performed these experiments because hypersensitivity to elevated sugar levels is a well-known, albeit poorly understood stress conditions for many cell wall-related mutants. It is therefore a standard assay in our laboratory. We are of course aware that it is not necessarily a proxy for a naturally occurring stress conditions but deem it useful to provide hints towards potential metabolic defects. Here, we simply note that this stress condition, artificial as it may be, reveals AGO10's function in maintaining phenotypic robustness.

I will suggest to the authors to stain starch with lugol, maybe *sgo1* mutants have some problem with sugar metabolism.

>> We thank the reviewer for this suggestion and have now included these experiments as Appendix Figure S3. However, we did not find any marked differences in Lugol staining. In addition to the standard staining of amyloplasts in the root cap, which is accomplished by very short staining of 30 secs, we also performed "overstaining" of two hours to reveal potential starch accumulation in the

root meristem above the QC but were not able to observe a difference between WT and *ago10* mutants.

Are genes of calvin cycle deregulated?

>> We have queried our RNA-seq dataset for indications deregulation of primary metabolism but could not observe any notable change of related genes.

This together with the miRNA165 expression levels would help to understand better the phenotype.

Optional:

Have the authors considered grafting experiments to understand the role of root and shoot in drought resistance of the *sgo1/ago10*?

>> This is an excellent suggestion that we will certainly pursue but we believe it is out of scope of the current study

Minor

- The authors suggest that AGO10 is required for the maintenance of miRNA 165/6 gradient. They show in figure 4 the expression localization of miRNA165a and AGO10 protein expression pattern but it is not clear which background are they using. Is Col 0 or *ago10*? If the reporter lines are in the mutant background, the authors should show the same reporter lines in the WT background.

>> We apologize for the confusion. We have now clearly indicated the background of the lines. Transcriptional reporters are in the Col-0 background, whereas the translational AGO10 reporter is in the *ago10-1* mutant background and shows near endogenous expression levels. In the accompanying study by Mirlohi et al., the behaviour of lines with expression levels lower and higher than that endogenous AGO10 (all in the *ago10-1* null background) is characterized in great detail. We now also include the pMIR165a:GFP expression in an AGO10 mutant background as Figure EV2D.

- Indicate genetic background of reporter lines in Fig 4B

>> The genetic background is now indicated in the figure legend.

- L112 Provide a full name for the RLP4, SDN1, SDN2 etc, when it's used for the first time.

>> We have included the full names.

- L206 Can the authors indicate if the mutation was dominant or recessive

>> The recessive nature of this allele is now indicated.

- L238-242 Shall the authors include the reference to figure 4B showing expression pattern of AGO10 in the stele?

>>The reviewer is correct, and we apologize for the omission. To our knowledge, stele expression of AGO10 was first shown by Iyer-Pascuzzi et al., 2011, which is now cited. It is also cited in the accompanying manuscript by Mirlohi et al.

- Higher-order combinatorial class III hd-zip mutants often show a triarch pattern in vascular tissue (three phloem poles). Is such phenotype ever observed in the sgo1?

>> In several hundred roots analyzed we have observed this phenotype exactly once and therefore have refrained from mentioning it in the manuscript.

- References with formatting error: Mahonen, 2006 (line 65, 167, 267,413), Mirlohi et al 2022 (line 158, 241, 369, 371), Ramachandran et al 2020 (line 299)...

>>We have corrected these citations

Referee #2:

In this manuscript, El Arbi et al. make a very valuable contribution to understanding the role of AGO10 in the control and the coordination of formative cell division and cell fate specification of the root vasculature. Also the manuscript is well written and the experiments are well justified, I have some comments to be addressed:

It would have been great if the figures would have captions under them to quickly recognize which figure is which.

Major comments:

- Line 111. I think it would be great if the authors elaborate a bit more on how the gene was discovered, the appearance is a bit vague in the text.

>>We initially studied RLP4 because it an evolutionarily conserved RLP and we have previously identified RLP44 to be involved in cell wall signalling. RLP4 seemed a good candidate for being involved in this process. In addition, RLP4 harbours a malectin-like domain which is speculated to be capable of cell wall binding. However, since there is no relationship to the study presented here and it might confuse the reader, we refrained from adding this information.

- Figure 2A. Also, the differences seem to be clear in the optical sections, it would be great to provide some z-stack reconstructions of a few cell layers to avoid the bias of sporadic appearance of EdU incorporation.

>>We thank the reviewer for the suggestion and now include 3D reconstructions of a median region as animated movies in the Appendix (Movies EV1 and 2).

- Figure 3E and F. I am not sure if it's easy to follow the "ectopic divisions" here. Please elaborate how these quantifications were done and what arrows are pointing at. At the same time, it would be great to see WT corresponding images and 3D reconstruction of TMO5 expression for a better comparison.

>>We have added information about the quantification and the meaning of the arrows in the text. In essence, we scored the presence of a pTMO5-positive cell file outside the xylem axis that could be

traced back to a precursor (arrow) as indicative of an ectopic division. In addition, we now provide 3D reconstructions of the pTMO5:GFP signal in Col-0 and *sgo1* background as Movies EV3 and EV4

- Figure 5B. I think it would be very informative to elaborate of what groups of genes the authors identify here, perhaps providing some GO terms at least.

>>We now provide an annotated list of DEGs as well as GO analysis in the Supplemental material. The most striking and consistent observation (to our eyes) was the upregulation of several JAZ protein genes in *sgo1*. As jasmonic acid has been linked to vascular development we intend to pursue this further and delineate whether the *ago10* phenotype partially acts through JA signalling. We now also include Figure EV3B, demonstrating that CK signalling is required for the increased cell numbers in *sgo1*, as expected.

- Figure 6A. The response to drought phenotype seems to be impressive in *sgo1* mutant. It would be important to discuss about the possible explanations for it in the Discussion section. Is it connected to more xylem formation, enhanced rate of formative divisions?

>>Here, we can only speculate as enhanced xylem formation is not always correlated with enhanced drought resistance. However, in the study by Bloch et al., it was observed that ABA induced HD-ZIP III downregulation resulted in increased drought resistance in seedling. We now cite this study in the discussion.

- Line 366. How do the miRNA165/6 and HD-ZIP III TFs behave in *ago10* background? It would be important to visualize it to support the statement that "in the absence of AGO10, miRNA165/6 can spread further from the endodermal source into the stele, curtailing HD-ZIP III expression."

>> see comment to reviewer 1: We agree with the reviewers that this is an important point which now has been comprehensively addressed in the accompanying manuscript by Mirlohi et al., demonstrating that PHB-GFP accumulation is indeed strongly reduced in *ago10* mutants.

We had initiated similar experiments and performed crosses between *ago10-1/sgo1* and a PHB-YPET line that became available. However, we were notified by the Voinnet laboratory that this line seems to show a HD-ZIP III loss-of-function phenotype and we could not confirm ectopic xylem that shows exclusively protoxylem-like patterning independent of the presence of AGO10. We therefore stopped these experiments and refer to the results of Mirlohi et al.

In addition, we have crossed the transcriptional miRNA reporter line *pMIR165b:GFP* with *sgo1* and observed that the spatial expression pattern was not affected by the absence of AGO10. In addition, qPCR with a primer pair directed against the *MIR166b* transcript revealed largely unaltered expression levels. These results are now included as Figure EV2D and E.

- Figure 5A. It would be very informative to provide optical sections of TCSn marker in WT and *sgo1* mutant to appreciate better the spatial distribution of cytokinin signalling, especially after BAP treatment.

>> We provide section at approximately 50 μm above the QC of the TCSn signal in Col-0 and *sgo1*

background, with and without BAP treatment, as Figure EV3A. In addition, we now include 3D-reconstructions of the pTCSn signal in Col-0 and *sgo1* as Movies EV5 and 6.

Minor comments:

- Please check the use of the definite articles throughout the text.
- Line 34. ".....acts by buffering cytokinin responses and restricting xylem differentiation."
- Line 36. Too many "show"
- Line 39. "Thus, AGO10 is required for the control of and coordination...."
- Line 63. Introduce the abbreviation for TFs.
- Line 81. Define TFs.
- Line 130. "We quantified the cell number close to the stem cell region." Please correct.

>> We thank the reviewer for spotting these mistakes, which are now corrected.

- Line 358. I assume the authors mean here rather xylem cell fate specification, as they don't focus on phloem phenotypes. Also, the divisions are affected in procambium, the authors focus primarily on xylem specification.

>>We see control of formative cell divisions and cell type specification as two separate processes, both affected by AGO10 and HD-Zip IIIs. Please note that divisions are affected in both procambium and xylem precursor cells. Moreover, under stress condition, even more xylem is specified in *ago10* mutants despite a reduction of ectopic cell divisions (Figure 6).

Referee #3:

El Arbi et al present a previously underexplored role for the Arabidopsis AGO10 gene in controlling root vascular development and phenotypic robustness. The work is performed at the highest level and the story is very cohesive. The notion that AGO10 controls periclinal/formative divisions, and limits phenotypic variability is interesting and important. I have only minor comments and am generally very supportive of publication.

1. What is unclear from the work is what the significance is of more/less phenotypic plasticity as conferred by different AGO10 levels. As presented, *ago10* mutants perform better in drought conditions, and roots have increased vascular cell numbers without compromising root length. Could the authors explore fitness penalties of the *ago10* mutation to better understand the role of phenotypic robustness/buffering provided by AGO10?

>>This is a very interesting point which is a part of a much larger complex of questions associated with phenotypic robustness/plasticity and its trade-offs. These questions are intensively discussed in the relevant literature and addressing this for our specific example would warrant a separate study, in our opinion. We therefore believe this to be out of scope of the current work where we can only conclude that AGO10 is required for the maintenance of phenotypic robustness. We note that the improved performance under drought conditions is not necessarily inconsistent with the notion of decreased phenotypic robustness as only few plants show an intermediate phenotype and around 50% of the individuals do not perform better than the WT.

2. While perhaps clear to specialists, Figure 6C would need more explanation for the uninitiated.

>>We apologize for not explaining this better. A kernel density plot shows the distribution of values in a dataset using a single continuous curve. It is thus similar to a histogram but “smoother” as there are no bins.

3. While the effect of AGO10 is predicted to be mediated by Class III HD-ZIP factors, the authors do not in fact show if/how these are affected. I would encourage the authors to show the accumulation pattern of at least one of these in the *ago10* mutant.

>> see comment to reviewers 1 and 2: We agree with the reviewers that this is an important point which now has been comprehensively addressed in the accompanying manuscript by Mirlohi et al., demonstrating that PHB-GFP indeed strongly reduced in *ago10* mutants.

We had initiated similar experiments and performed crosses between *ago10-1/sgo1* and a PHB-YPET line that became available. However, we were notified by the Voinnet laboratory that this line seems to show a HD-ZIP III loss-of-function phenotype and we could confirm ectopic xylem that shows exclusively protoxylem-like patterning independent of the presence of Ago10. We therefore stopped these experiments and refer to the results of Mirlohi et al.

Dear Sebastian,

We have now received re-review reports from two referees, which I have included below. As you will see, you have addressed their concerns satisfactorily. Before I can finally accept the manuscript though, there are some remaining editorial points which need to be addressed. In this regard would you please:

- add up to five keywords,
- rename the conflict of interest statement as the "Disclosure and competing interests statement",
- remove the AC/CrediT section from the text,
- include figure callouts for Fig. 5D, and amend callouts for EV figures to Fig. EV1-EV5, (instead of just EV1-EV5), remove callouts for Figures S4C and S6,
- complete the general info table in the author checklist,
- remove the dataset legend from the manuscript file and upload it as a separate tab in the Excel file,
- include page numbers in the appendix table of contents,
- provide a URL for dataset GSE218342 in the data availability statement,
- label data description summaries at the end of all legends as 'Data Information',
- as figure 5b does not contain a microscopy image, remove the scale bar related information in the figure legend,
- define the annotated p values ***/**/* in the legend of figures EV 2c; EV 5c; as appropriate,
- indicate the statistical test used for data analysis in the legends of figure 5b; EV 2b-c; EV 5c,
- correct the discrepancy between stated p values in figure and figure legend in Fig. 5b; EV 4a; EV 5a-b,
- define box plot parameters in the legend of figure EV 3b,
- state sample sizes in the legends of figures EV 2c; EV 3b,
- provide a numbered scale bar for the expression analysis present in figure 5c,
- define scale bars in figures 4a-c, g; EV 1a; EV 3a (right), and include scale bars and their definition for figure 6f and EV 3a (left),
- define the white arrow in the legend of figure 4c,
- correct the section order, placing the main and EV figure legends after the references, and
- Zip each movie file with its corresponding legend,

We include a synopsis of the paper (see <http://emboj.emboPress.org/>). Please provide me with a general summary image, statement and 3-5 bullet points that capture the key findings of the paper.

I am looking forward to receiving your revised manuscript.

EMBO Press is an editorially independent publishing platform for the development of EMBO scientific publications.

Best wishes,

William

William Teale, PhD
Editor
The EMBO Journal
w.teale@embojournal.org

See also figure legend guidelines: <https://www.emboPress.org/page/journal/14602075/authorguide#figureformat>

- a point-by-point response to the referees' comments, with a detailed description of the changes made (as a word file).
- a word file of the manuscript text.
- individual production quality figure files (one file per figure)
- a complete author checklist, which you can download from our author guidelines

(<https://www.embopress.org/page/journal/14602075/authorguide>).
- Expanded View files (replacing Supplementary Information)
Please see out instructions to authors
<https://www.embopress.org/page/journal/14602075/authorguide#expandedview>

We realize that it is difficult to revise to a specific deadline. In the interest of protecting the conceptual advance provided by the work, we recommend a revision within 3 months (12th May 2024). Please discuss the revision progress ahead of this time with the editor if you require more time to complete the revisions. Use the link below to submit your revision:

Referee #2:

The authors' revision has significantly enhanced the manuscript, and I have no further comments to add at this time.

Referee #3:

The authors have considered my points, and offer good explanations. I am satisfied with this revision.

Dear Editor, dear William,

thank you very much for your and the reviewers' positive evaluation of our "**AGO10 controls cell fate specification and formative cell divisions in the Arabidopsis root**". We have addressed your editorial points and hope you will find this version acceptable for publication.

Looking forward to hearing from you.

With best wishes and on behalf of all co-authors,

Sebastian Wolf

Point-by-point response

Referee #2:

The authors' revision has significantly enhanced the manuscript, and I have no further comments to add at this time.

Referee #3:

The authors have considered my points, and offer good explanations. I am satisfied with this revision.

>> We thank both reviewers for their constructive evaluation of our manuscript.

Editorial points

- add up to five keywords,
- rename the conflict of interest statement as the "Disclosure and competing interests statement",
- remove the AC/CrediT section from the text,
- include figure callouts for Fig. 5D, and amend callouts for EV figures to Fig. EV1-EV5, (instead of just EV1-EV5), remove callouts for Figures S4C and S6,
- complete the general info table in the author checklist,
- remove the dataset legend from the manuscript file and upload it as a separate tab in the Excel file,

- include page numbers in the appendix table of contents,
- provide a URL for dataset GSE218342 in the data availability statement,
- label data description summaries at the end of all legends as 'Data Information',
- as figure 5b does not contain a microscopy image, remove the scale bar related information in the figure legend,
- define the annotated p values *****/**/*** in the legend of figures EV 2c; EV 5c; as appropriate,
- indicate the statistical test used for data analysis in the legends of figure 5b; EV 2b-c; EV 5c,
- correct the discrepancy between stated p values in figure and figure legend in Fig. 5b; EV 4a; EV 5a-b,
- define box plot parameters in the legend of figure EV 3b,
- state sample sizes in the legends of figures EV 2c; EV 3b,
- provide a numbered scale bar for the expression analysis present in figure 5c,
- define scale bars in figures 4a-c, g; EV 1a; EV 3a (right), and include scale bars and their definition for figure 6f and EV 3a (left),
- define the white arrow in the legend of figure 4c,
- correct the section order, placing the main and EV figure legends after the references, and
- Zip each movie file with its corresponding legend,

We include a synopsis of the paper (see <http://emboj.embopress.org/>). Please provide me with a general summary image, statement and 3-5 bullet points that capture the key findings of the paper.

>>We have addressed all issues as suggested with the exception of “- provide a numbered scale bar for the expression analysis present in figure 5c,” as the colour code is relative here, i.e. the extremes of the spectrum denote highest and lowest expression within one row (representing one gene across the conditions).

Dear Sebastian,

I am pleased to inform you that your manuscript has been accepted for publication in the EMBO Journal.

Congratulations! I am really glad to see this study in EMBO Journal.

Best wishes,

William

William Teale, PhD
Editor
The EMBO Journal
w.teale@embojournal.org
